



# Wind tunnel investigations of an individual pitch control strategy for wind farm power optimization

Franz V. Mühle[1,*], Florian M. Heckmeier[2,*], Filippo Campagnolo[1], and Christian Breitsamter[2]

[1]Chair of Wind Energy, Technical University of Munich, Boltzmannstr. 15, 85748 Garching bei München, Germany
[2]Chair of Aerodynamics and Fluid Mechanics, Technical University of Munich, Boltzmannstr. 15, 85748 Garching bei München, Germany
[*]These authors contributed equally to this work.

**Correspondence:** Franz Mühle (franz.muehle@tum.de), Florian M. Heckmeier (florian.heckmeier@aer.mw.tum.de)

**Abstract.**

This article presents the results of an experimental wind tunnel study, which investigates a new control strategy named Helix. The Helix control employs individual pitch control for sinusoidally varying yaw and tilt moments to induce an additional rotational component in the wake, aiming to enhance wake mixing. The experiments are conducted in a closed-loop wind tunnel under low turbulence conditions to emphasize wake effects. Highly sensorized model wind turbines with control capabilities similar to full-scale machines are employed in a two-turbine setup to assess wake recovery potential and explore loads on both upstream and downstream turbines. In a single-turbine study detailed wake measurements are carried out using a fast-response five-hole pressure probe. The results demonstrate a significant improvement in energy content within the wake, with distinct peaks for clockwise and counter-clockwise movements at Strouhal numbers of approximately 0.45. Both upstream and downstream turbine dynamic equivalent loads increase when applying the Helix control. The time-averaged wake flow streamwise velocity and RMS-value reveal a faster wake recovery for actuated cases in comparison to the baseline. Phase-locked results with azimuthal position display a leapfrogging behavior in the baseline case in contrast to the actuated cases, where distorted shedding structures in longitudinal direction are observed due to a changed thrust coefficient and an accompanying lateral vortex shedding location. Additionally, phase-locked results with the additional frequency reveal a tip vortex meandering, which enhances faster wake recovery. Comparing the Helix cases with clockwise and counter-clockwise rotations, the latter exhibits slightly higher gains and faster wake recovery. This difference is attributed to the Helix additional rotational component acting either in the same or opposite direction as the wake rotation. Overall, both Helix cases exhibit significantly faster wake recovery compared to the baseline, indicating the potential of this technique for improved wind farm control.

## 1 Introduction

The growing demand for renewable energy sources has led to the increased deployment of wind turbines in many parts of the world. However, the power output of wind turbines can be significantly impacted by the complex flow dynamics in their wakes. The reduction in the incoming wind velocity and increased turbulence caused by the wake can reduce the efficiency of downstream turbines, limiting the overall power output of a wind farm. Therefore, understanding and controlling the wake





behaviour is critical to enhance the efficiency of wind farms and increase their power output. Therefore, wind farm control is
identified to be important to fulfil the main challenges in wind energy science stated by Veers et al. (2019). The last decades vast
research was directed to understanding the wake and applying different techniques to control it with various aims, from power
optimization to load reduction. A comprehensive review of recent wind farm flow control strategies was published by Meyers
et al. (2022).

Lately, wake mixing techniques, which aims at enhancing recovery by perturbing the wake at specific frequencies attracted
the attention of the research community. In the literature, several control strategies which aim to influence the turbulent mixing
of wind turbine wakes are presented. An example for such a control strategy is to periodically change the yaw misalignment
with respect to the incoming wind, which results into bending perturbations in the wake that lead to a faster recovery. This
was investigated numerically by Kimura et al., who showed potential for this technique. In another study Munters and Meyers
(2018a) showed that a dynamic turbine excitation is triggering wake meandering. Moreover, they showed that this effect is de-
creased for increasing turbulence intensity. Dynamic induction control (DIC), where the induction factor is dynamically varied
by a combination of pitch and torque control is another possibility to enhance wake mixing. In this way axial perturbations
are generated in the wake. This control strategy was investigated numerically by Munters and Meyers (2017) and Yılmaz
and Meyers (2018) and also experimentally by Frederik et al. (2020b). However, this studies show, that the application of this
technique results in increased dynamic loads on the downstream turbines. Furthermore, Munters and Meyers (2018b) suggest
that it is only efficient applying such a technique to first-row turbines in low turbulence conditions. The loads on the upstream
and downstream turbine should be limited by another way of dynamically influencing the wake by mixing, which is termed the
Helix approach being introduced by Frederik et al. (2020a). In the Helix approach, individual pitch control (IPC) is used to si-
nusoidally change the blade pitch resulting in a variation of the fixed-frame tilt and yaw moments. Consequently, an additional
excitation of the wake is introduced. In their CFD study, Frederik et al. (2020a) explain the concept of the Helix technique
and show the meandering of the wake and the potential for enhanced wake mixing. In another numerical study, Frederik and
van Wingerden (2022) investigate the influence of DIC and Helix on the tower and blade loads. They show that both wake
mixing techniques increase the loads of upstream and downstream turbines, whereas the Helix approach has a higher effect
on the turbine blades than on the tower. The authors conclude, that despite the increased loads, wake mixing techniques are
an option for full-scale applications. van den Berg et al. (2022) conduct a CFD analysis to evaluate the effect of the Helix
approach applied to floating wind turbines. Their results suggest, that the Helix control strategy can even be more efficient
when applied to floating turbines compared to bottom fixed machines. In a more recent study, Taschner et al. (2023) perform
large-eddy simulations of a two-turbine wind farm and check the effect of a various pitch amplitudes. They find an increasing
mixing effect with increasing pitch angles showing no saturation in the investigated pitch angle range, which reaches up to
pitch angles of $6°$. Furthermore, they show the occurence of increasing loads when increasing the pitch amplitudes, suggesting
an optimal operation of the Helix as a trade off between power gain and loading. Since most of these studies focus on the nu-
merical investigation of the Helix approach, a thorough experimental verification can be barely found to date. For this reason,
this study investigates the potential of the Helix approach experimentally in a wind tunnel (W/T) and should give a detailed
insight in the wake aerodynamics.





This present study is guided by two research questions. Firstly, the effects of the Helix approach are examined focusing on both the turbine level observations and the fluid flow in the turbine wake. This will allow an assessment whether Helix can enhance the entrainment of incoming wind flow and improve the efficiency of power extraction in wind farms. Secondly, the identification of the underlying mechanisms in the wake that lead to faster wake recovery are investigated. This will provide insights into the complex flow dynamics behind wind turbine wakes and the role that Helix plays in modifying these dynamics. Understanding these flow mechanisms is critical to optimize the use of Helix in wind farm design and operation. By addressing these research questions, a contribution to the development of more effective control strategies for wind turbines and a deeper understanding of the flow dynamics in wind turbine wakes should be achieved.

To address the research questions outlined above, a methodology that combines wind tunnel (WT) experiments with detailed wake analyses is applied. At first, turbine level experiments to study the effects of the Helix approach on the wind turbines in tandem configuration are conducted. This involves installing sensors on the wind turbine to measure its performance and analysing the data to assess the impact of the Helix strategy on power output and experienced loads. This allows an assessment of the performance of the wind turbine under different Helix control conditions and identify any improvements resulting from the application of Helix. Finally, a detailed wake analysis to identify the mechanisms in the wake leading to faster wake recovery is carried out. This involves analyzing the mean flow field as well as phase-locked flow structures behind the wind turbine, in order to provide an understanding of the effects of Helix on the performance of wind turbines and the complex flow dynamics in their wakes.

In the following sections, the control strategy (section 2) - the Helix approach - and the experimental setup (see section 3) will be described in detail, with emphasis on the advanced techniques employed for data acquisition and processing. In the section on the wind tunnel results, section 4, the turbine level results will be presented and analyzed first, identifying interesting excitation strategies for further investigation. The wake study results will be presented in two parts - a time-averaged wake analysis and detailed phase-locked studies of the tip vortex area. Here, the discussion deals with the interpretation of the results and shows implications for wind turbine design and operation. Finally, the conclusions and outlook in section 5 will summarize the key findings and their significance, as well as offering an outlook for further research and development of control strategies in wind turbine technology.

## 2  Individual Pitch Control: the Helix approach

In order to optimize the power output of wind farms, it is essential to understand the aerodynamics in the wind turbine wake. The low speed region governing the wake is the reason that multiple successive wind turbines need to be separated with a certain spatial distance between each other. By controlling the upstream turbine wake, e.g. introducing additional instabilities into the flow, synergetic effects on the entire wind turbine farm can be achieved. Hence, controlling and influencing the wake recovery has a huge potential for wind farm power optimization. Basically, the flow behind a wind turbine is defined by the continuous sheet of vorticity which rolls up to two bigger vortices, the tip vortex and the root vortex. The root vortex imposes a





rotary movement on the wind turbine wake that is counter rotating to the turbine rotation. The tip vortices build a helical system in the wake of the wind turbine. The wake aerodynamics can be separated into three main phenomena, the vortex shedding, the tip-vortex pairwise instability – also known as leapfrogging instability –, and the turbulent mixing (see Lignarolo et al. (2015),

Sarmast et al. (2014) and Sørensen (2011)). If it is possible to influence the tip-vortex breakdown, and thereby increase the net entrainment of kinetic energy, the wake can be re-energized and therefore, recovers earlier. The predominant instability mode in wind turbine wakes is the mutual-inductance instability, where adjacent helical filaments influence each other and start to roll up which results in the leapfrogging phenomenon (Lignarolo et al., 2015).

A control strategy, taking these mechanisms into account was introduced by Frederik et al. (2020a). In the so-called *Helix*
*approach*, the wind turbine blades experience a dynamic individual pitch control/excitation (DIPC) resulting in a variation of the fixed-frame tilt and yaw moments and a dynamically variation of the direction of the thrust force. Thereby, they show a meandering of the wake in either clockwise (CW) or counter clockwise (CCW) direction, depending on the slightly out of sync excitation frequency. In their proof of concept, Frederik et al. (2020a) show the potential of this technique and demonstrate that if it is applied, a faster wake recovery is detected. However, in the literature, merely results of computational simulations
are present so far. For this reason, in this article, the potential of the Helix approach is experimentally investigated in a wind tunnel (W/T) and should give insight in the wake aerodynamics.

In the following, the dynamic individual pitch control strategy, which was implemented in the model wind turbines to achieve a sinusoidal variation of the fixed-frame tilt and yaw moments and consequently a dynamic variation in the thrust force direction, will be explained.

The turbine was controlled by individual pitch control (IPC), where each blade experiences the same pitch excitation at the same azimuthal position. This is shown schematically for an excitation with the rotational frequency $f_r = \omega_r/60 = 1P$ in figure 1a). This is in contrast to collective pitch control (CPC), where each blade experiences the same pitch excitation at the same time.

In case of a clean lab flow (low turbulence intensity $Tu < 0.5\%$, no shear), a sinusoidal variation of the fixed-frame tilt and
yaw moments is assumed to be achieved by a sinusoidal variation of pitch amplitudes. A proof of this assumption will be given in section 4.1. As a result, in the presented study, the pitch angle $\beta_b(t)$ amplitudes served as input for the controller. The blades are individually controlled with a sinusoidal excitation with a frequency $f_\beta = f_r \pm f_e$ which is out of sync with the rotational frequency $f_r = 1P$. The additional excitation frequency $f_e$ is either added or subtracted to the rotational frequency, leading to the CCW or CW wake meandering, respectively.

Blade pitch signals for the two cases, $f_\beta < 1P$ (Helix 0.82) and $f_\beta > 1P$ (Helix 1.18), are visualized in figure 1b) for a fixed blade pitch offset $\beta_{b,0} = 0°$.

The following equation shows the transient blade pitch angle $\beta_b(t)$ for each of the three blades $b = \{1,\ 2,\ 3\}$:

$$\beta_b(t) = \beta_{b,0} + \overbrace{\hat{\beta} \cdot \sin\left(2\pi \underbrace{(f_r \pm f_e)}_{f_\beta} t + \theta_{b,0}\right)}^{\Delta\beta} \tag{1}$$



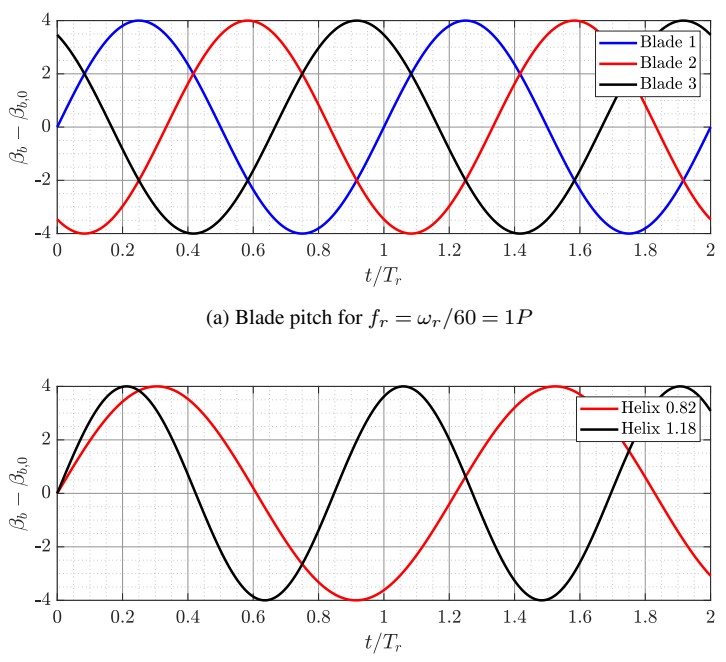

(a) Blade pitch for $f_r = \omega_r/60 = 1P$

(b) Comparison of the actuated cases for blade 1

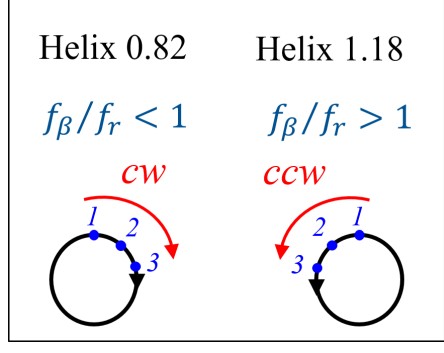

(c) Helix direction explanation

**Figure 1.** a) Blade pitch for all three blades for $f_r = 1P$, b) blade pitch of blade 1 for $f_r < 1P$ and $f_r > 1P$, with $\hat{\beta} = 4°$ as a function of the normalized time $t/T_r$, and c) explanation of the Helix directions

Here, the blade pitch excitation amplitude is denoted as $\hat{\beta}$, the respective blade pitch offset $\beta_{b,0}$, and the azimuthal blade

position $\theta_{b,0} = -(2\pi/3) \cdot (b-1)$ at $t = 0\ s$. Since, the *G1 model* is three-bladed, the azimuthal blade positions are $\theta_{1,0} = 0°$, $\theta_{2,0} = -120°$ and $\theta_{2,0} = -240°$.

In case $f_e$ is subtracted, $f_\beta/f_r$ will be smaller than one (Helix 0.82). Consequently, the additional rotation will be in the same rotational direction as the rotor, as one pitch period is longer than one rotation period of the rotor. This is schematically shown in figure 1c) for a clock-wise rotating turbine, where the blue line indicates the turbine rotation and the red line indicates

the additional rotation for three pitch periods. Contrary, in the Helix 1.18 case (adding $f_e$), the pitch period is shorter than the rotor rotation and thus the additional rotation is in opposite direction of the rotor rotation.

To be able to better compare the Helix excitation methodology with other dynamic control strategies, the additional frequency $f_e$ can also be expressed in terms of the Strouhal number, correlating the additional excitation frequency $f_e$ with the blockage-corrected inflow velocity $U_{\infty,corr}$. It is calculated as $St_{add} = \frac{f_e D}{U_{\infty,corr}}$, where $D$ is the rotor diameter. In this way the Strouhal

numbers will be identical if the same $f_e$ is added or subtracted, the difference will be indicated by specifying a CW or CCW rotational direction. In the article, values for both notations $f_\beta/f_r$ and $St$ will be provided.





## 3 Experimental Setup

In the following section, the experimental setup of the conducted wind turbine model tests in the boundary layer wind tunnel at TUM are described. The section is separated giving details on the model wind turbines, the wind tunnel itself, the measurement
stages and the applied intrusive measurement equipment, a fast-response five-hole probe.

### 3.1 Model wind turbines

For the wind tunnel tests, two identically scaled wind turbine models are arranged in a tandem configuration. The rotor diameter of the machines, depicted as G1 models in the following, is $D = 1.1\ m$. A detailed description of the G1s is given by Bottasso and Campagnolo (2021). Nevertheless, a brief summary of the applied sensors and controls is given in the following: The
operating rotor speed of the three bladed G1 is $840\ rpm$. The blades are manufactured out of unidirectional carbon fibre. Each blade is equipped with an individual pitch actuator and a built-in relative encoder measuring the pitch angle. The rotor azimuthal position $\theta$ is detected via an optical encoder. Furthermore, a torque-meter can measure the torque in front of the torque generator. Moreover, full-bridge strain gauges are used to measure the torque and the two out-of-plane bending moments on the rotating shaft, between the rotor and the front bearing. Similarly, sensors located at the tower base allow measuring
the fore-aft and side-side bending moments therein. Finally, nacelle nodding and yawing bending moments are derived by projecting the shaft rotating loads into a fixed axis system. The G1 model is controlled with an industrial real-time Bachmann M1 controller. The controller is connected to a supervisory PC, where all settings and recordings concerning the wind turbine can be done. The control algorithm for the Helix strategy is implemented in the turbine controller. While the rpm is kept fixed at an optimum value throughout the experiments, the demanded blade pitch is varied according to Eq. 1.

### 3.2 Wind tunnel

The wind tunnel experiments are carried out in the closed-loop (Göttingen-type) low-speed wind tunnel C (W/T-C) of the Chair of Aerodynamics and Fluid Mechanics of the Technical University of Munich (TUM-AER). Besides power measurements of the wind turbine itself, transient fast-response five-hole pressure probe measurements in the wake of the wind turbine model are intended. The W/T-C closed test section has a size of $1.8 \times 2.7 \times 21.0\ m^3$ (height x width x length). The ceiling of the wind
tunnel is adjustable to minimize the pressure gradient in the longitudinal direction. Turbulence intensity lies below $Tu < 0.5\%$. In general, in order to simulate the atmospheric boundary layer as seen in the real world case, it is possible to equip the wind tunnel with a vortex generator and roughness elements. Nevertheless, in this study, the boundary layer instrumentation is omitted in order to emphasize the aerodynamic effects of the Helix actuation while operating the wind tunnel with low turbulence intensity. The blockage ratio $\alpha = A_{turbine}/A_{windtunnel}$ in the experiment lies at $\alpha = 0.2$, which is rather high. To account for
this blockage effect, the wind tunnel free-stream velocity was adjusted to $U_\infty \approx 5.3\ m/s$, which is lower than the rated wind speed of the G1 models (Bottasso and Campagnolo, 2021). As a result of the blockage, the fluid is however accelerated, resulting in a higher velocity experienced by the turbines. An estimation of the Rotor Effective Wind Speed (REWS) for the upstream G1, done as described in Campagnolo et al. (2022), revealed a blockage-corrected free-stream velocity $U_{\infty,corr} \approx 5.9\ m/s$,



which correlates to the rated wind speed of the model wind turbine. Consequently, the turbine is operated at a tip-speed ratio
of approx. $\lambda = 8.2$ . In the analysis of the conducted measurements in section 4, the presence of low turbulence inflow and the
rather high blockage should be considered when interpreting the results and especially when applying the learnings to realistic
wind turbine flows. Consequently, the data has to be seen as tendencies and not providing insight in realistic power gains of
full scale wind turbine tandem configurations.

### 3.3   Measurement stages

The measurement campaign is divided into two scenarios/stages:

In the first stage (see Figure 2 a)), two turbines are placed in line with a longitudinal distance of 5D (see figure 2 a)). The
upstream turbine is actuated following the control strategy introduced in section 2. The downstream turbine acts purely as a
sensor, providing an integral insight on the energy content in the wake and thus the recovery behind the first turbine. With this
setup, a vast variety of different actuation frequencies and amplitudes can be tested in order to identify the control settings
characterized by the most prominent and promising effects.

As pre-tests, a range of pitch amplitudes $\hat{\beta} = (1° : 1° : 5°)$ were tested to investigate the capabilities of the pitch system,
and specifically to assess the largest achievable amplitude given the ultimate currents and pitch rate of the available actuators.
As a result, within the presented investigation, the Helix actuation implemented on the upstream turbine (actuated G1) is
achieved with a pitch amplitude $\hat{\beta} = 4°$. This control is expected to produce the greatest possible effects on the wake shed by
the machine, without exceeding the actuation limits.

In this study, tests were therefore conducted by solely changing the additional pitch excitation frequency $f_e$, which is con-
trolled by setting a desired pitch frequency $f_\beta$ in the Bachmann M1 controller. For the test with the two turbine setup $f_\beta$
was varied within the range $10.1 : 0.3 : 17.9\ Hz$. The upstream turbine is operated with a constant rotational frequency of
$f_r = 840\ rpm/60 = 14\ Hz$ and an optimal pitch offset of $\beta_0 = 0.4°$. These controls result in a non-dimensional actuation fre-
quency of $f_\beta/f_r = (0.72 : 0.02 : 1.28)$, which corresponds to a Strouhal number $St_{add}$ within the range $0 : 0.052 : 0.73$ both
in CW and CCW direction. The downstream turbine (sensor G1) serves as a sensor and is down-rated to $f_r = 750\ rpm$ and
has a pitch offset of $\beta_0 = 0°$.

The results of these experiments were used to identify the two control set-points ($f_\beta/f_r > 1$ and $f_\beta/f_r < 1$) capable of trigger-
ing the fastest wake recovery, i.e. the ones inducing the largest increases in the power produced by the downstream machine.

In the second stage (see figure 2 b)), the downstream turbine is removed from the wind tunnel. To better understand the
underlying physics of the Helix approach, the wake shed by the actuated G1 is traversed with a fast-response five-hole pressure
probe (FRAP, see section 3.4). Specifically, the wake is captured at hub height, and along a x-y-plane spanning only the right
side of the turbine (see figure 2a). The wake measurements are performed for three different actuation cases: the reference one,
characterized by $\hat{\beta} = 0°$; the "Helix 0.82"/CW, with $f_\beta/f_r = 0.82$ and $St_{add} = 0.47$; the "Helix 1.18"/CCW, with $f_\beta/f_r =$
1.18 and $St_{add} = 0.47$. These two Helix cases were the ones producing the largest increase in power production for the
downstream G1, as discussed in section 4.1.

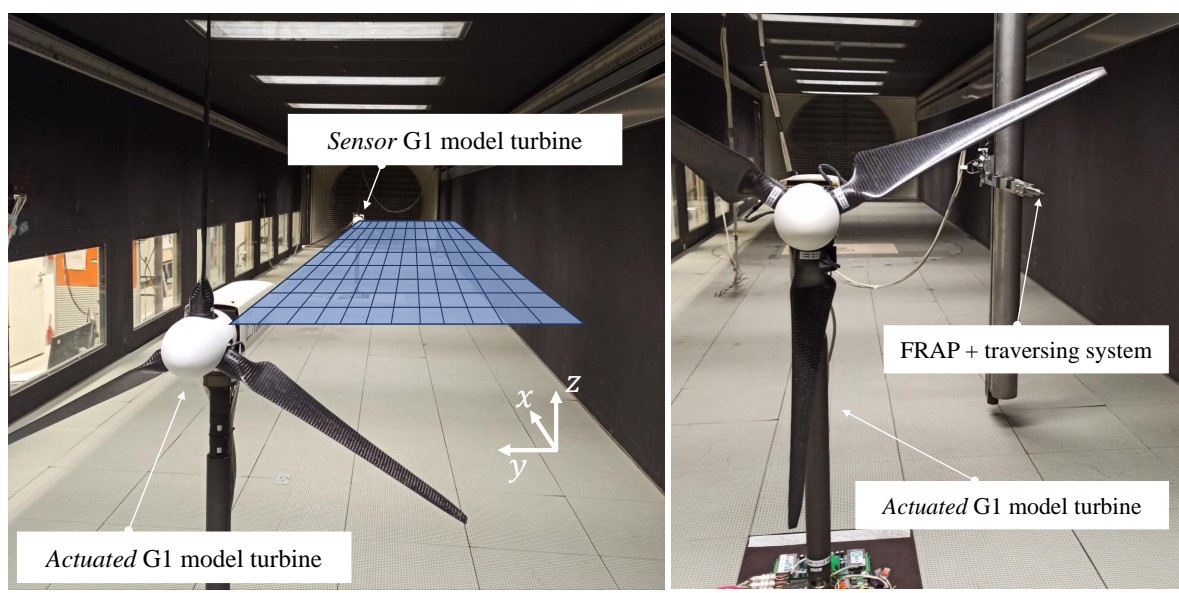

(a) 1. stage setup: two turbines                          (b) 2. stage setup: one turbine + FRAP

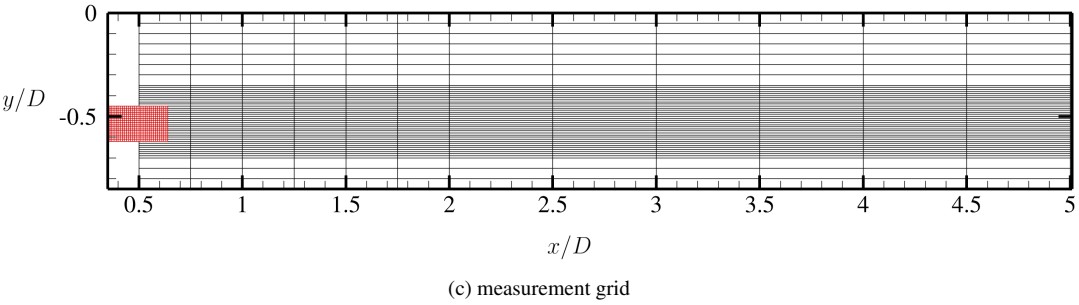

(c) measurement grid

**Figure 2.** Wind tunnel setup for the two measurement stages: a) two G1 wind turbine models and a schematic representation of the FRAP measurement grid and b) one G1 turbine and the FRAP in TUM-AER W/T-C c) measurement grids for wake measurements (black) and refined wake measurements (red).



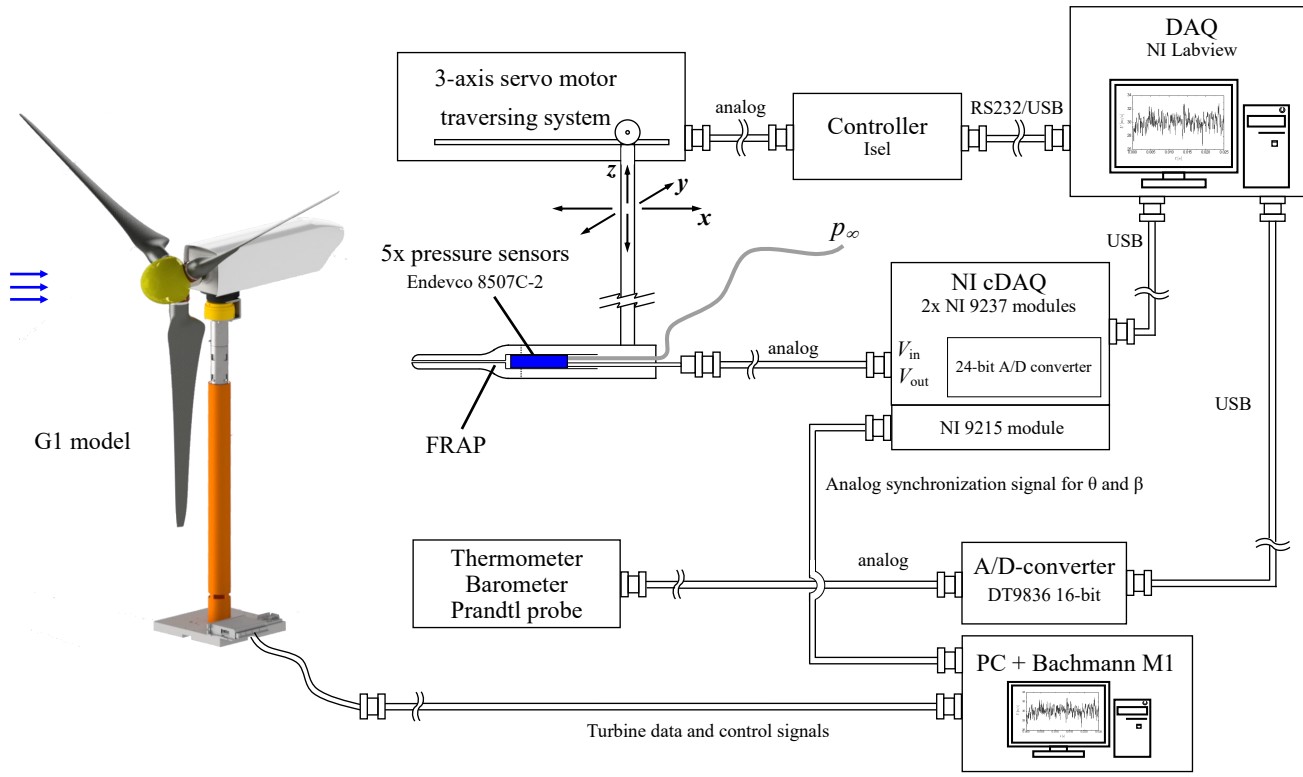

**Figure 3.** Schematic fast-response five-hole probe and G1 wind turbine model measurement setup for TUM-AER W/T-C

The measurement grid is shown in figure 2c). Radial mappings at hub height were taken at 13 downstream locations $x/D = [0.5 : 0.25 : 2.0, 2.0 : 1.0, 5.0]$, with $x/D = 0$ the position of the rotor disk. For each radial mapping, measurements were taken

at unevenly spaced locations $y/D = [0 : 0.05 : 0.35, 0.35 : 0.01 : 0.7, 0.7 : 0.05 : 0.85]$. In addition, a refinement region ($y/D = 0.46 : 0.01 : 0.62$ and $x/D = 0.35 : 0.01 : 0.64$) in the very near wake of the turbine, highlighted in red colour in figure 2c, is used to further investigate the vortex shedding mechanism .

In figure 3, a schematic visualization of the W/T measurement setup for the FRAP wake measurements is given. The FRAP is mounted on a three-axes traversing system. The servo traversing motors are controlled with an Isel servo-controller managed

by LabVIEW. Moreover, the ambient air properties (static pressure $p_s$ and temperature $T_\infty$) and the dynamic pressure of the undisturbed inflow $q_\infty$ are measured. The free-stream velocity is monitored with a Prandtl probe that is installed, at hub height, approx. two diameters upstream of the actuated G1 turbine. Data is acquired with a Data Translation 16-bit A/D-converter. The FRAP pressure data are sampled with NI cDAQ 9237 full-bridge modules operated at a sampling frequency of $f_s = 10\ kHz$. In order to account for aliasing problems, a low-pass filter below the Nyquist-Shannon frequency $f_{LP} \leq f_s/2$ is applied in the

NI measurement card. The measurement time is set to $t_s = 40.0\ s$. In addition to the acquisition of the five pressure sensor signals, the azimuthal position $\theta$ and the blade pitch $\beta$ of the first blade of the actuated G1 are digitized with a NI-9215 analog



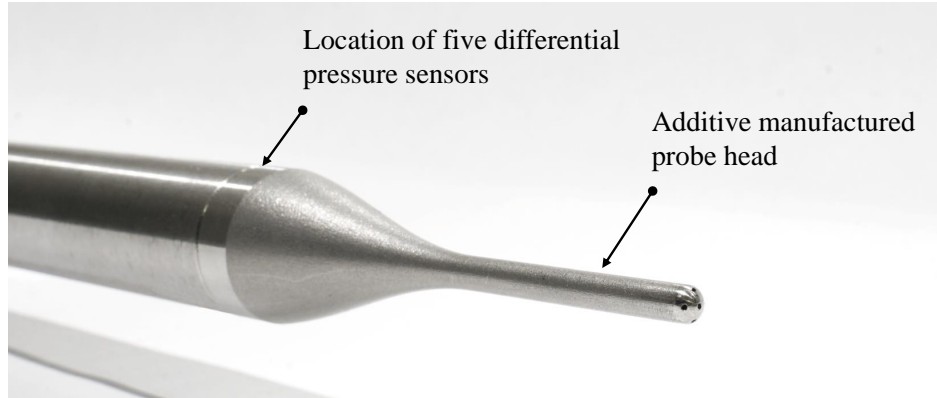

**Figure 4.** Fast-response five-hole probe equipped with piezo-resistive sensors

input cDAQ module. This allows the FRAP and wind turbine measurements to be synchronized through a dedicated post-processing routine. Moreover, it allows the use of phase-locking techniques for a transient investigation of the full wake of the wind turbine. A similar approach has been followed by Mühle et al. (2020) for studying the effects of winglets on the tip vortex interaction within the wake shed by a wind turbine model.

### 3.4 Fast-Response Five-Hole Pressure Probe

In the following, detailed information on the characteristics and calibration of the used fast-response pressure probe (FRAP) for the measurements in the turbine wake is given:

The FRAP probe head is manufactured in an additive manufacturing process. Further design details, like the hemispheric probe tip shape are post-manufactured in a cutting step. Five Meggitt Endevco 8507C-2 differential piezo-resistive pressure sensors are equipped inside the probe head and are pressurized with the ambient pressure outside of the wind tunnel as reference pressure, Heckmeier et al. (2019). The reference pressure lines of the differential pressure transducers are merged in a manifold. The sensors are connected to the NI cDAQ 9174-chassis with its inserted NI cDAQ 9237 data acquisition cards. The acquired data is transmitted to the NI LabVIEW controlled computer.

The FRAP is calibrated for its spatial/aerodynamic and its temporal/transient characteristics. Some details on the respective calibration process are given in the following. The spatial and temporal characteristics are assumed to be decoupled and both calibrations are performed separately. A detailed description of the underlying theory can be found in Heckmeier et al. (2019); Heckmeier and Breitsamter (2020); Heckmeier (2022). By measuring the five pressure signals and setting them into relation in the reconstruction process, the flow properties at the probe tip can be deduced. In the spatial/aerodynamic calibration, pressure data sets for different angle combinations at various flow velocities are acquired in a free-jet calibration wind tunnel. Thereby, various steady angle combinations are set and calibrated, resulting in a maximum reconstruction range for the FRAP of up to $\pm 60°$. The pressures are processed within four non-dimensional calibration coefficients $b_1$, $b_2$, $A_t$, and $A_s$, which are used in a local interpolation approach. During the reconstruction, the measured data $p_{1,T}, p_{2,T}, ..., p_{5,T}$ are post-processed. Here, the



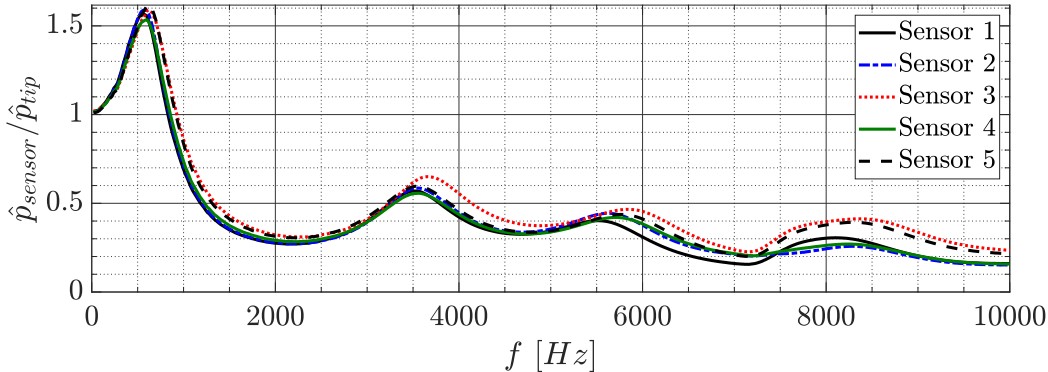

**Figure 5.** Attenuation of the probe tip signal in the line-cavity system as a result of the transfer function determination in the frequency test rig

subscript $T$ denominates the values at the unknown test point $T$. The non-dimensional coefficients $b_{1,T}$ and $b_{2,T}$ are calculated then. A local-least square interpolation determines the quantities $A_{t,T}$, $A_{s,T}$ and $\alpha_T$, $\beta_T$ as functions of $f(b_{1,T}, b_{2,T})$. Here, $\alpha$ and $\beta$ are the two flow angles. In the last reconstruction and interpolation step, the Mach number $Ma$ can be calculated. In the temporal/transient calibration, the line-cavity system inside the probe is investigated and characterized. The acoustic system is mainly dominated by two effects, resonance and attenuation. Bergh and Tijdeman (1965) analytically formulated a recursive solution for small disturbances in such systems. The attenuation and phase shift for the acoustic wave propagation inside a single-tube system is formulated as the complex ratio $H(\omega) = P_{sensor}(\omega)/P_{tip}(\omega)$ and is called transfer function (TF). The transfer function of the FRAP in this study is experimentally determined in a frequency test-rig, where the probe is inserted in a closed acoustic chamber. Sinusoidal signals are emitted by a speaker and recorded by the sensors inside the probe and a reference sensor located next to the probe tip at specified frequency steps ($\Delta f = 20\ Hz$). Hence, the transfer function is obtained. The temporal calibration is performed for frequencies up to $10\ kHz$, Heckmeier et al. (2019). Figure 5 depicts the attenuation and resonance effects inside the five line-cavity systems of the FRAP.

In the temporal reconstruction process, the measured time series pressure data in the unknown flow is transformed into frequency domain with a fast Fourier-transformation (FFT) and the TF is applied on the measured data:

$$P_{sensor}(\omega) = H(\omega)P_{tip}(\omega) \tag{2}$$

By applying an inverse FFT, the reconstructed pressure data at the probe tip is available and can be further processed with the spatial reconstruction as described above.

To sum up, table 1 lists some important pressure probe calibration and geometric dimension properties. For the reconstruction of the flow-field properties, the measured pressures are post-processed taking the calibration data of both, temporal and spatial,



**Table 1.** Five-hole probe properties

| | |
|---|---|
| Tip diameter | $3\ mm$ |
| Channel diameter | $\leq 1\ mm$ |
| Sensor type | differential, piezo-resistive |
| Sensor gauge pressure range | $2\ psig$ |
| Sensor diameter | $2.3\ mm$ |
| Spatial/angular calibration | $\pm 60°$ |
| Temporal calibration | $10\ kHz$ |

calibrations into account. A high reconstruction accuracy below $0.2°$ in both flow angles and $0.1\ m/s$ in the reconstructed
velocity can be achieved, as shown in Heckmeier and Breitsamter (2020).

## 4 Results

In the following section, the results of the experimental measurement campaign will be presented and discussed. Firstly, the
results on turbine level will be analysed to see how the turbine is influenced by applying the Helix technique and to identify,
how the turbine performance is affecting the wake and its recovery. Secondly, the results of the wake will be presented. Here,
the section starts with a time-averaged analysis and is followed by results from phase-locked investigation using the rotor
azimuth position and the additional Helix frequency for phase locking.

### 4.1 Turbine data

The turbine data presented in this chapter was gathered by the sensors located onboard of the G1s, which are described in more
detail in Section 3.


Figure 6 shows the nacelle nodding moment of the upstream turbine, corrected by the gravity-induced load. Specifically,
the figure depicts the moment recorded during 14 rotor revolutions (equivalent to one second of measurement) for the baseline
case, the *Helix 0.82/CW* case and the *Helix 1.18/CCW* case. The yawing moment would show a similar behaviour, just offset
by 90°and would not reveal new insight. For this reason it is not reported here. The graph confirms the expectation, i.e. that the
implemented pitch actuation leads to the desired behaviour of the nodding and yawing moments. The moment for the baseline
case is indeed only fluctuating slightly due to mechanical vibrations and the not null turbulence intensity of the wind tunnel
inflow. The nodding moment for the two *Helix* cases, instead, show a very clear sinusoidal variation at the desired frequency,
i.e. 2.5 Hz.

This is confirmed by the Fourier transformation single sided amplitude spectra of the nodding and yawing moments in the
fixed-frame coordinate system. In addition to the peak at the rotational frequency $f_r = 14\ Hz$, the Helix cases also experience



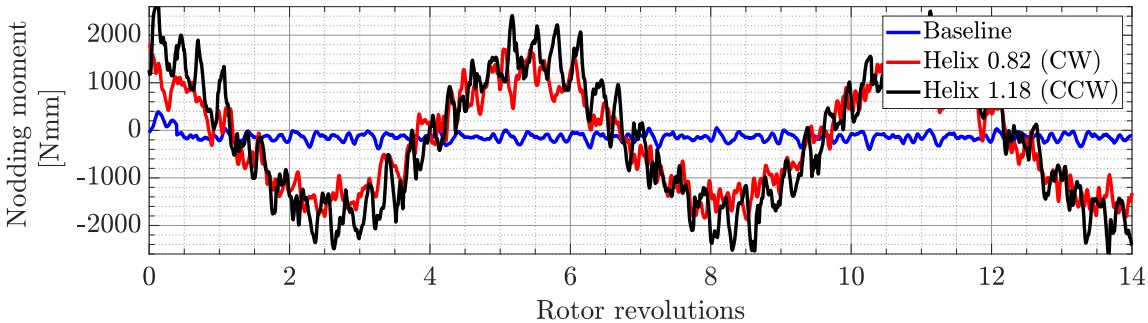

**Figure 6.** Fixed-frame nodding moment (time domain) for baseline (blue), Helix 0.82 (red), and Helix 1.18 (black).

a significant peak at the additional frequency $f_e \approx 0.18 f_r = 2.5\ Hz$ which is more than one order of magnitude bigger than the rotational frequency events.

The extracted powers of both wind turbines can be seen as an integral metric for the turbine performance. The summation of the two powers shows the overall performance of the whole setup. Figure 7 depicts the results of the power investigations by changing the actuation frequency of the first turbine in the range of $f_\beta / f_r = (0.72 : 0.02 : 1.28)$. The extracted powers $P$ are normalized by the power of the upstream/actuated turbine in the baseline scenario $P^*$, where no actuation is present and the pitch is kept constant to $\beta = \beta_0 \neq f(t)$. In general, the extracted power of the actuated turbine decreases for all test frequencies, since it is operated in a non-optimal operating point. With an increasing actuation frequency, this effect is also increasing. The behavior in terms of extracted power of the downstream/sensor turbine shows a remarkable trend. The curves are not perfectly mirrored along an axis in the $1P$ frequency, but clearly, two distinct local maxima can be identified at $f_\beta / f_r < 1$ and $f_\beta / f_r > 1$. As already discussed in Frederik et al. (2020a), these two peaks are located at the identical additional excitation frequency of $f_e = 0.18 f_r$ referring to identical Strouhal numbers $St_{add} \approx 0.45$. Moreover, it can be seen that the trend of decreasing power with higher excitation is not applying for the downstream turbine, where the effect is vice-versa, what is suggesting higher available power in the wake with a faster pitch excitation of the front turbine. When looking at the total power of the two-turbine setup, which is presented by the black line in Figure 7, one can see that the opposite power behaviour of the upstream and downstream turbine are almost evened out showing an nearly symmetrical behaviour for clockwise and counter-clockwise rotation. In the following, the wake investigations will focus on these two peaks. The corresponding scenarios are named according to the helical movement of the wake in either clockwise (CW) $f_\beta / f_r < 1.0$ or counter-clockwise direction (CCW) $f_\beta / f_r > 1.0$. The identified pitch frequencies are $(f_\beta / f_r)|_{CW} = 0.82$ and $(f_\beta / f_r)|_{CCW} = 1.18$, the two cases are hereafter referred to as *Helix 0.82* and *Helix 1.18*, respectively.

Figure 8 shows the thrust coefficient $C_T$ of the actuated upstream turbine for different additional excitation frequencies. The trend for $C_T$ is opposite to the one of the available power: with increasing actuation frequencies, $C_T$ is increasing, whereas a slight ditch can be observed at $f_\beta / f_r = 1.0$ $(St_{add} = 0)$. However, towards the very high actuation frequencies $f_\beta > 1.2 f_r$ this increase seems to have reached a maximum and is again decreasing slightly.




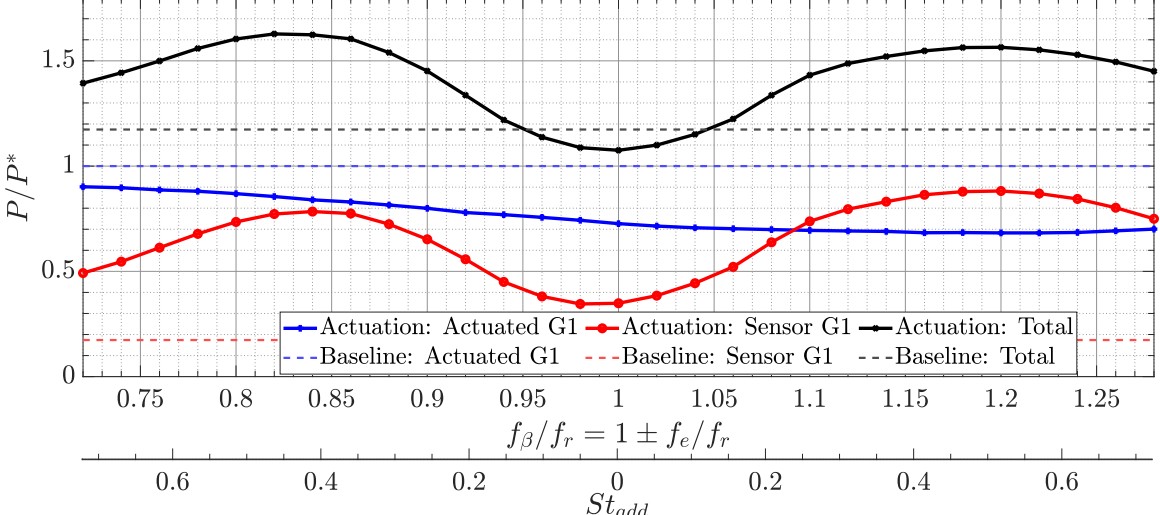

**Figure 7.** Extracted power of the upstream/actuated turbine, downstream/sensor turbine and combined turbine array, for changing pitch frequencies $f_\beta/f_r = (0.72 : 0.02 : 1.28)$ compared to the baseline case without any actuation $f_\beta = 0$ (dashed line).

Nevertheless, the higher $C_T$ values at faster actuation would suggest a larger velocity deficit in the wake. However, the available power for the downstream turbine has a higher peak for $f_\beta/f_r > 1.0$ than $f_\beta f_r < 1.0$. Consequently, the differences in power peaks of the sensor turbine for $f_\beta/f_r \neq 1.0$ must come from mixing mechanisms in the wake, which might be caused by the opposite rotational direction of the additional wake rotation. Moreover, as sen in Figure 8, the $P/P^*$ graph of the sensor turbine has two distinct peaks at almost identical Strouhal numbers for CW and CCW rotational direction. These two peaks

are not present for the actuated upstream turbine, which is barely Strouhal number dependent. Additionally, it can be assumed that this effect is influenced by wake mixing mechanisms and is not influences by the performance of the upstream turbine.

   In addition, the application of the IPC for the Helix control technique is without a doubt increasing the duty cycles of the pitch mechanism compared to the non actuated baseline case, as the IPC control is demanding a constant operation of the pitch motors to achieve the sinusoidal pitch motion. Figure 9 instead reports the Damage Equivalent Loads (DELs) at the

rotating shaft (HubRotDEL, a) and at the nacelle (HubFixedDEL, b) for both G1s, which were computed as follows. First, load signals were filtered above six times the rotor frequency ($6 \times$ Rev) in order to remove high-frequency load components. Signals demodulation was then adopted for correcting the nacelle loads of their $1 \times$ Rev harmonic component due to a slight inertial and aerodynamic imbalance of the rotor. Once removed the imbalance-induced spurious harmonic, the loads were then projected back to the rotating frame to obtain corrected values of the rotating shaft loads. Moreover, DELs were obtained by projecting

the corresponding two orthogonal bending moments on the direction associated to the maximum DEL. Furthermore, the figure depicts normalized DEL, obtained dividing the DELs by $1/2\rho\pi R^3 V_{\mathrm{REWS}}^2$. Finally, data pertaining to the case $f_\beta/f_r = 1.0$ ($St_{add} = 0$) are not included in the plots. In this condition the pitch actuation resembles the expected behavior of IPC for load reduction (Bossanyi, 2003), but without the loads feedback. It is not worthwhile, therefore, to analyze the corresponding DELs.

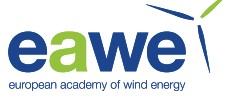
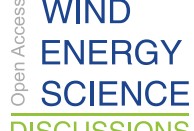


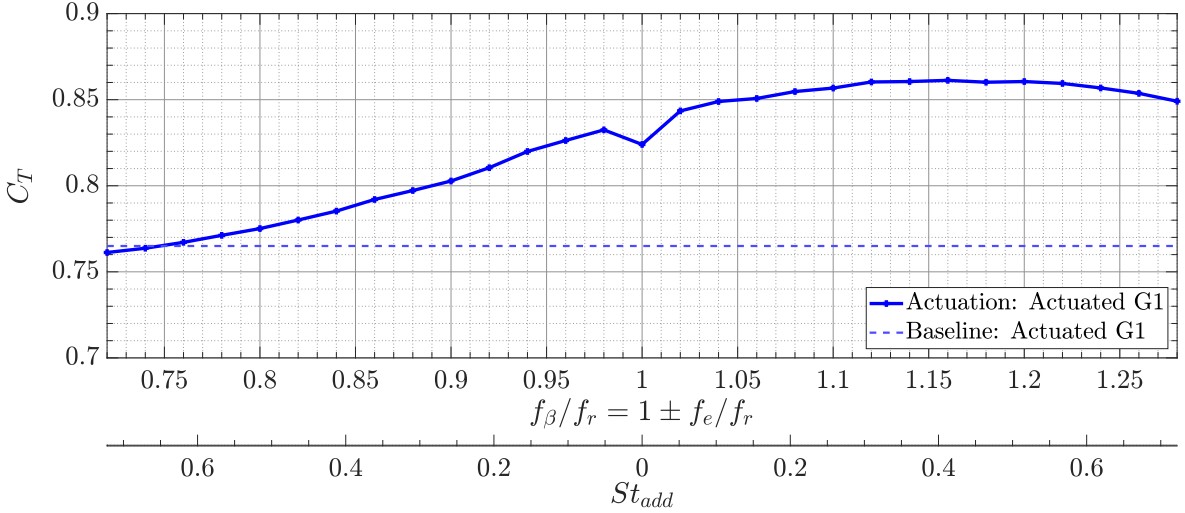

**Figure 8.** CT of the upstream turbine for changing pitch frequencies $f_\beta/f_r = (0.72 : 0.02 : 1.28)$ compared to the baseline case without any actuation $f_\beta = 0$ (dashed line).

The depicted data, reported as function of $f_\beta/f_r = (0.72 : 0.02 : 1.28)$, highlights that the loads are increasing, both for
the upstream and downstream turbine and for both load channels, if the front turbine implements the Helix. For the upstream turbine the loads are showing a trend similar to the $C_T$ one, as they are increasing with ascending actuation frequency. For the sensor turbine, the loads do not show a dependency towards the actuation frequency but towards the Strouhal number. They almost show an identical shape for a CW and a CCW actuation. For lower Strouhal numbers the loads are larger. At the optimal Strouhal number ($St_{add} \approx 0.45$) they are almost at the level of the reference case. The dependency on the Strouhal number is
similar to the behaviour of the available power behind the actuated turbine. However, for the loads investigation, there are no distinct peaks comparable to the ones observed for $P/P^*$ of the downstream turbine.

Summarizing, a sinusoidal individual pitch actuation is causing a sinusoidal variation of the nodding and yawing moment with a strong energy content at the additional rotation frequency. The hub loads for the actuated and sensor turbine are increasing in case the Helix control is applied, whereas the actuated turbine shows a larger load increase than the sensor turbine.
If the turbine is controlled with the Helix technique, the power is decreasing with ascending additional actuation frequency. However, the power readings of the sensor turbine show very strong gains of the available power behind the actuated turbine with two distinct local maxima at almost identical Strouhal numbers $St_{add}$. These cases with the maximum extracted power ($(f_\beta/f_r)|_{CW} = 0.82$ and $(f_\beta/f_r)|_{CCW} = 1.18$ at $St_{add} \approx 0.45$) are chosen for the wake investigations in the next section and are compared to the wake of the reference case.



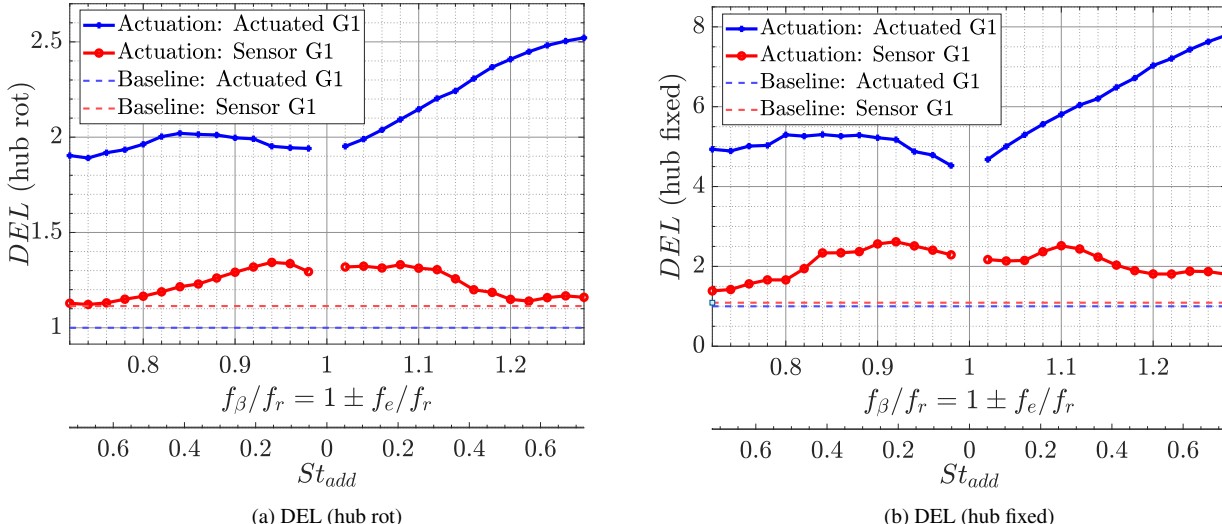

(a) DEL (hub rot)  (b) DEL (hub fixed)

**Figure 9.** Normalized DELs of the upstream/actuated turbine and downstream/sensor turbine for a) rotating hub and b) fixed hub, for changing pitch frequencies $f_\beta/f_r = (0.72:0.02:1.28)$ compared to the baseline case without any actuation $f_\beta = 0$ (dashed line).

## 4.2 Wake data

The following section covers the main results of the measurements conducted with the FRAP in the wind turbine wake for the reference case and the two actuated cases $(f_\beta/f_r)|_{CW} = 0.82$ and $(f_\beta/f_r)|_{CCW} = 1.18$ at $St_{add} \approx 0.45$. Hence, the downstream turbine is removed from the W/T (2. stage setup) and the probe is traversed according to the measurement grid described in section 3. All measurement are conducted at hub height $z/D = 0$.

### 4.2.1 Time-averaged wake flow

Firstly, the time averaged flow field will be presented and discussed. In figure 10, contour plots of the normalized time-averaged streamwise velocity component $\bar{u}/U_\infty$ for the baseline, Helix 0.82, and Helix 1.18 cases are shown. In accordance with the findings in the extracted power observations, the wake velocity has recovered earlier for the actuated cases. This effect is already noticeable at a short distance behind the rotor where a weaker wake border is present for the actuated scenarios and is getting more pronounced at larger downstream distances. However, small differences in the wake shape can be seen between the two actuated cases. The Helix 1.18 streamwise velocities at lateral positions close to the centerline recover earlier and show higher wake velocities. While the baseline case wake provides a sharper border in the tip vortex region, it starts to show entrainment at downstream distances around $x/D = 2$. Nevertheless, a clear separation between the low and high energetic flow fields is still present at $x/D = 5$. In the actuated cases, already at the most upstream measurement location $x/D = 0.5$, the tip vortex region is quite broad and no distinct sharp wake border can be identified. Consequently, the wake is not shielded from the outer, high energetic flow and entrainment of energy starts much earlier as in the baseline case. This leads to a faster

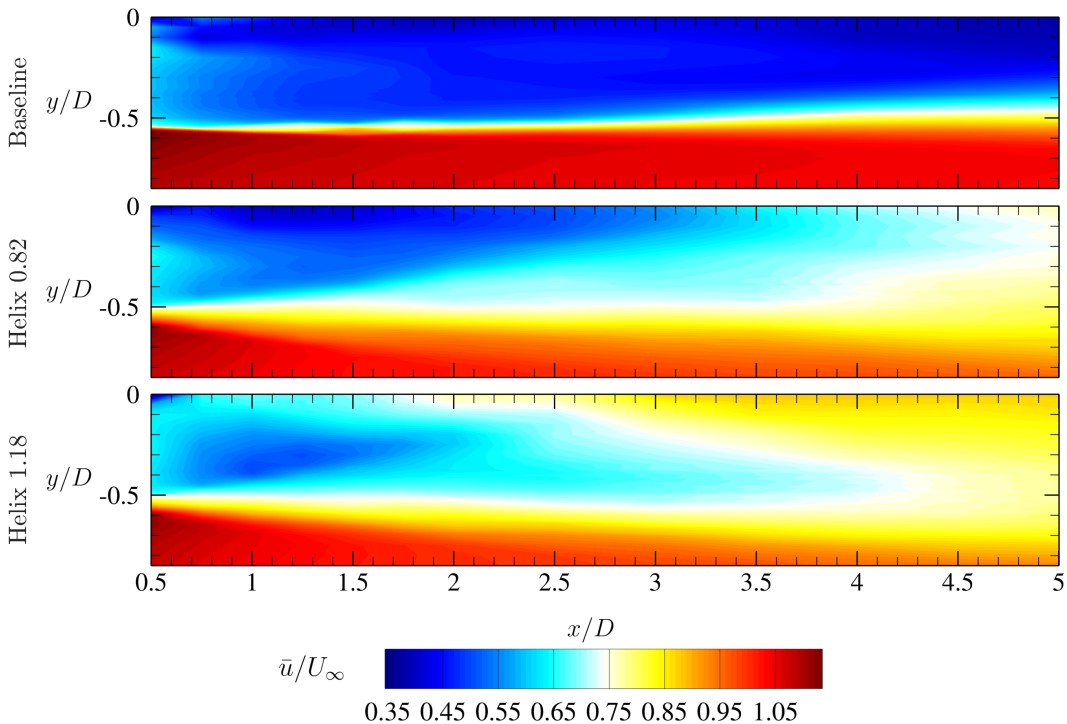

**Figure 10.** Contour plot of the normalized time-averaged streamwise velocity component $\bar{u}/U_\infty$ for the Baseline case and the two actuated cases Helix 0.82 and Helix 1.18.

recovery, which can be confirmed by the higher velocities in the wake region. Both actuated cases show streamwise velocities above $\bar{u}/U_\infty > 0.7$ at the former location of the downstream turbine (sensor G1 in 1. stage setup) at $x/D = 5.0$, confirming the two distinct power peaks observed in Figure 7.

In order to determine the instability mechanisms that are dominant and the location they occur, figure 11 shows contour plots of the normalized time-averaged RMS-values of the streamwise fluctuations $u'_{rms}/U_\infty$ for the three cases. For the baseline case, the highest fluctuations occur in the region where the tip vortices move downstream and separate the wake. Between $x/D = (1.0, \ 2.0)$, the onset of a mixing process can be detected, and hence, leapfrogging is expected. In contrast to the flow patterns in the baseline case, in the actuated cases, increased fluctuation contents are already detectable in the wake. These

fluctuations enable an entrainment of the energy-containing flow from outside into the wake. Thereby, mixing is promoted and an earlier recovery of the wake is visible.

     As a next step, in order to discover the governing mechanisms in the actuated cases, kinetic energy spectra are calculated for various locations in the wake of the turbine. Figure 12 shows the spectra at four locations $P1 = (x/D, y/D)_1 = (0.5, -0.55)$, $P2 = (2.0, -0.55)$, $P3 = (0.5, -0.15)$ and $P4 = (2.0, -0.15)$. As expected, the baseline spectra solely experience peaks at the

rotational frequency $f_r = 14 \ Hz$ and its higher harmonics. The peak magnitudes decrease with increasing distance from the turbine. In the more inward locations inside the wake at $y/D = -0.15$, these peaks are barely detectable. Looking at the spectra





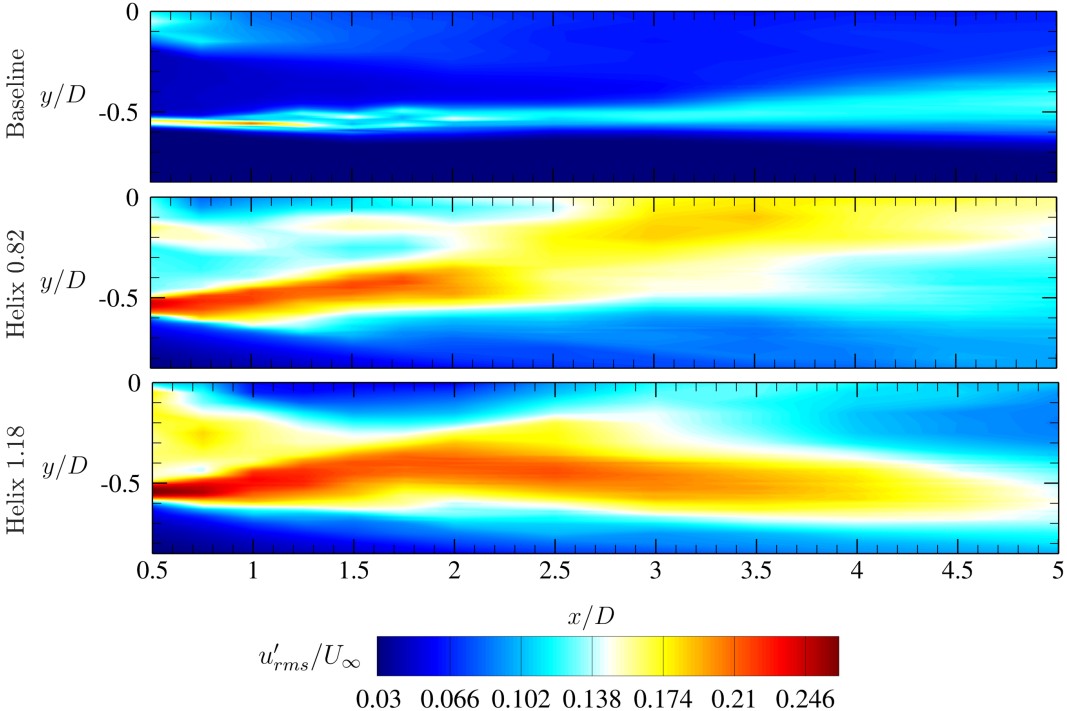

**Figure 11.** Contour plot of the normalized time-averaged RMS-values of the streamwise fluctuations $u'_{rms}/U_\infty$ for the Baseline case and the two actuated cases Helix 0.82 and Helix 1.18.

of the actuated cases, the system is governed by multiple higher harmonic peaks, based on the first harmonic of the additional excitation frequency $f_e = 2.5 \ Hz$. As already seen in the turbine hub moments, the system is governed by the excitation frequency $f_e$ instead of the rotational frequency $f_r$ as seen in the baseline flow. Especially in the low bandwidth, additional

energy is added for the actuated cases. Together with the already shown velocity plots in figure 10, it can be concluded that low bandwidth content due to the additional out of sync actuation is introduced in the entire wake $|y/D| < 0.5$ leading to earlier entrainment and faster wake recovery.

In addition, transient flow data are analyzed with respect to phase-locked information.

### 4.2.2 Wake flow analysis: results phase-locked with the rotor azimuthal position

In this section, a more detailed analysis of the transient wind turbine wake is conducted using phase-locked post-processed data obtained from the FRAP measurements and the azimuthal blade position signal. Figure 13 schematically depicts the principle of the phase-locking process. Each rotation with a rotational period $T_r$ is equally divided into $N$ segments. In this schematic figure, the full rotation is exemplarily divided into $N = 12$ segments $\Delta\theta = 360°/N = 30°$. All data points inside a segment and in corresponding segments of same phase (same color) in the upcoming rotational periods are gathered and

averaged to a single value, hence, the phase-locked value for this segment. In the wake measurements, the azimuthal position





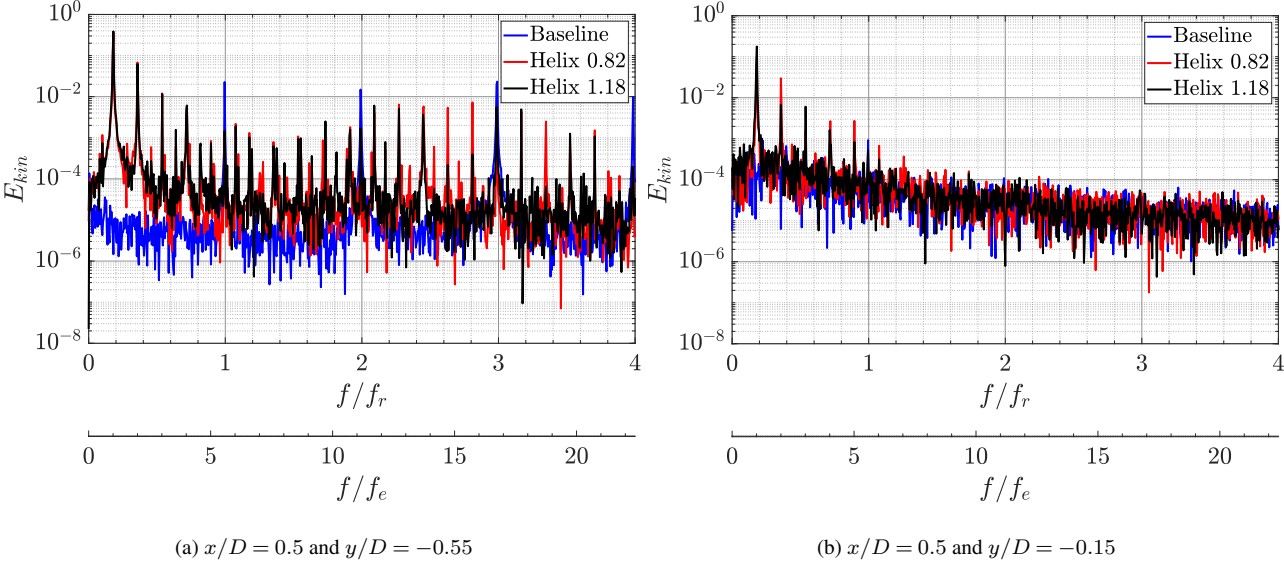

(a) $x/D = 0.5$ and $y/D = -0.55$          (b) $x/D = 0.5$ and $y/D = -0.15$

**Figure 12.** Kinetic energy spectra $E_{kin}$ at various measurement locations in the turbine wake at $x/D = 0.5$ and $y/D = \{-0.15, -0.55\}$.

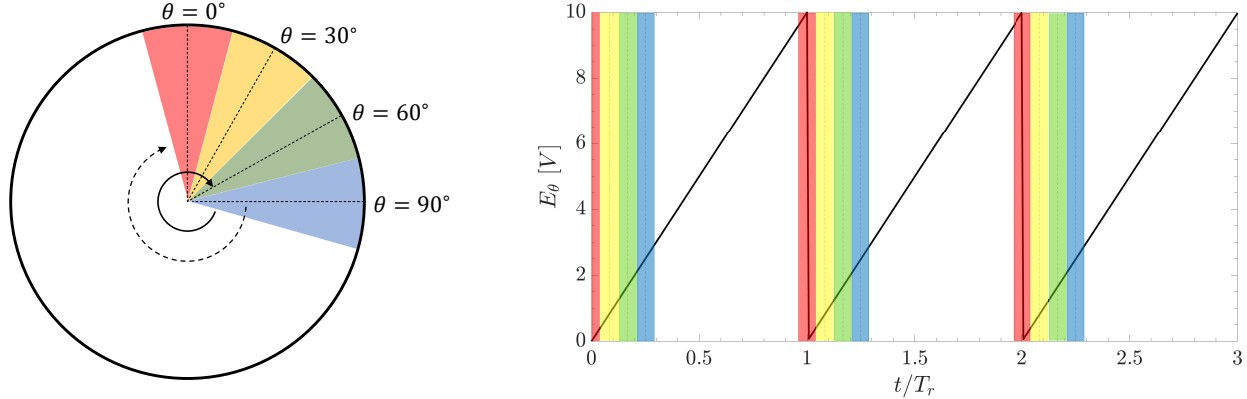

**Figure 13.** Phase-locking principle visualized by the rotational location of the blade $\theta$ and the acquired analog output signal voltage $E_\theta$.

$/\theta$ is processed in form of a saw tooth analog signal, where $E_\theta = 0\ V$ and $E_\theta = 10\ V$ correspond to $\theta = 0°$ and $\theta = 360°$, respectively. So the FRAP measurements can be conducted without any further trigger mechanism and are fully synchronized in the post-processing steps via the shown phase-locking procedure.

In the following analysis, one rotational period was divided into 120 segments. Thus, each phase-locked value maps data of
a segment of $\Delta\theta = 3°$.





It has to be noted that the results present the phase-locking with the rotational period $T_r$. Since the actuation of the blade pitch is slightly out of sync, the results cannot fully show a transient behavior as it really occurs at the blade! Solely, a phase-locked averaged investigation is possible. For the baseline case, since no blade pitch actuation is active, the phase-locked results represent a realistic visualization of the flow over one rotor revolution.

Figure 14 represents the data basis for the discussion of the rotor azimuth phase-averaged results of the turbine wake. Data was gathered at four downstream line locations $x/D = \{0.5, \ 1.0, \ 2.0, \ 5.0\}$ and are shown for the phase-averaged rotation $\theta = 0 - 360°$. Furthermore, in order to link to the time averaged results from section 4.2.1, time averaged line plots are also shown in the figure. For the sake of consistency, the axes are switched to have a matching y-axis with the phase-locked contour plots. The phase-averaged results for $u'_{rms,\theta}/U_\infty$ will be discussed in the following, as they contain the necessary information to explain the mechanisms in the wake. The index $()_\theta$ indicates the phase-averaged property.

Closely behind the rotor, the baseline case experiences three distinct, separated peaks at $y/D \approx -0.55$ due to the three tip vortices per revolution. Furthermore, the fluctuation level resulting from the root vortex system near the centerline $y/D \approx -0.05$ is increased and higher fluctuations are detected. In the baseline case, between $x/D = (1.0, \ 2.0)$, the leapfrogging instability mechanism occurs, which is shown by a representation of the normalized phase-locked RMS-values of the streamwise fluctuations $u'_{rms,\theta}/U_\infty$. At $x/D = 1.0$, a clear separation of three distinct vortices per revolution can be still detected. The phase difference between the occurrence of the vortices is still approximately $\Delta\theta = 120°$, as expected for a three-bladed turbine. However, a first inward movement is seen, and hence, the instability and interaction between the helical vortices is introduced. In more downstream positions, the roll up process of multiple vortices starts and results in an almost complete merging of the three vortices to a single structure at $x/D = 2.0$. At $x/D = 2.0$ the individual vortices merged completely into a coherent structure.

When discussing the velocity fluctuations in the line plots in the top row of Figure 14, higher levels of perturbations can be seen throughout all lateral positions $y/D$ for the Helix cases for all investigated downstream distances $x/D$. Thus, more mixing is already present in the very near wake for the actuated cases. It is therefore assumed that no conventional instability mechanism, like the leapfrogging mechanism, can be accounted for the wake recovery. This observation is enhanced by the contour plots for the actuated cases. At the nearest downstream line location at $x/D = 0.5$, a band of high fluctuations with separated peaks ($\Delta\theta \approx 120°$) is detected for both Helix cases. These structures are not showing coherent vortex filaments as in the baseline case. They are already merged at the downstream location $x/D = 1.0$. At $x/D = 5.0$ the flow for the actuated cased is almost uniform along the phase-locked time. Compared to the Helix 0.82 case, the Helix 1.18 case shows a more uniform and higher fluctuation level throughout the full revolution. As it can be seen in the more downstream contour figure, the additional fluctuations provoke an earlier mixing and entrainment of higher velocity flow from the outer area leading to a more uniform fluctuation level at $x/D = 5.0$.

In order to understand the flow physics for the actuated cases better and to explain the band of high fluctuations directly behind the rotor, a further refinement at locations closer to the turbine is performed. A refined measurement grid in the region between $y/D = (-0.45, -0.62)$ and $x/D = (0.35, 0.64)$ (shown in red in Figure 2 c) for both actuated cases is defined and

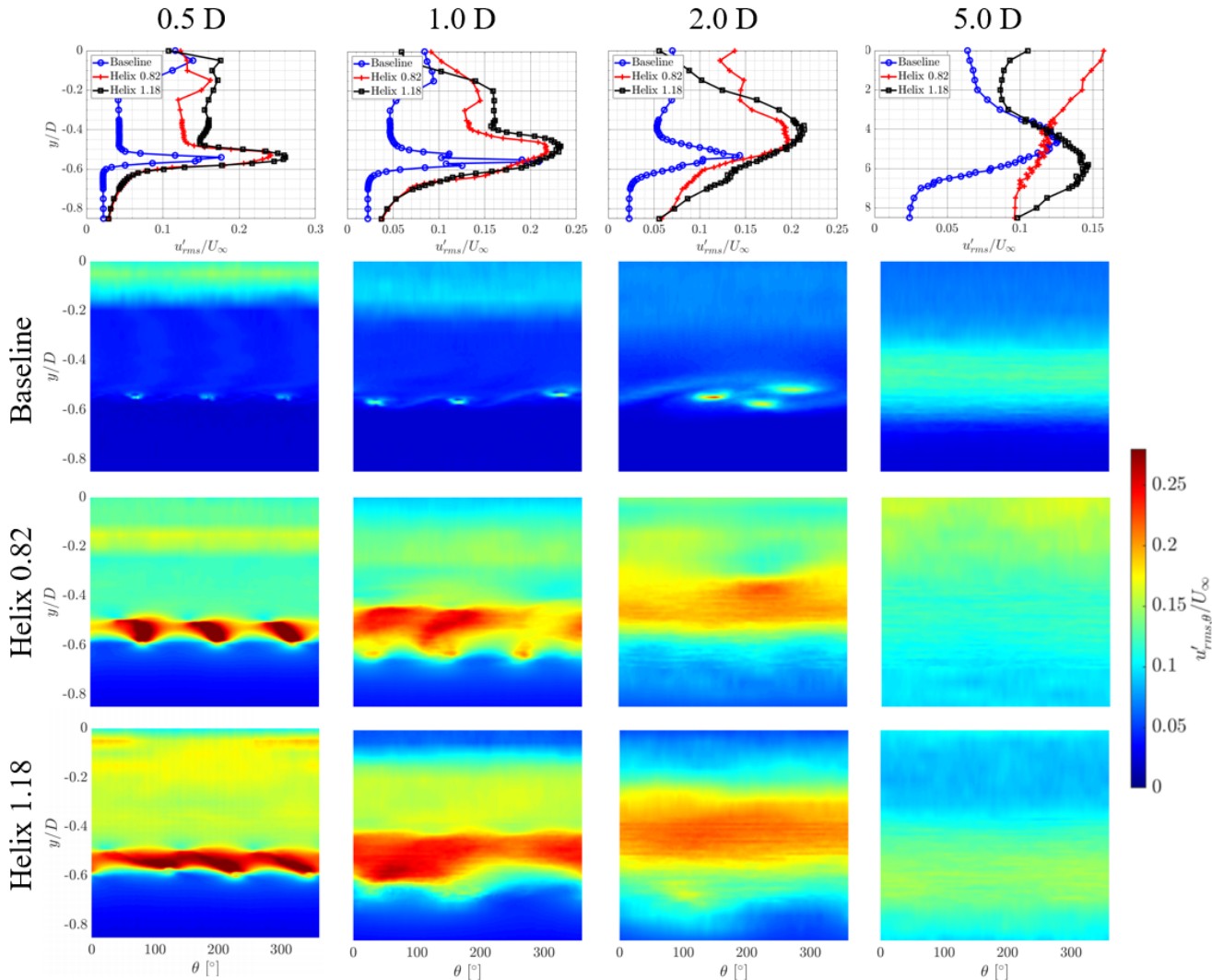

**Figure 14.** Normalized phase-locked RMS-values of the streamwise fluctuations $u'_{rms,\theta}/U_\infty$ at multiple downstream positions $x/D = \{0.5, 1.0, 2.0, 5.0\}$.





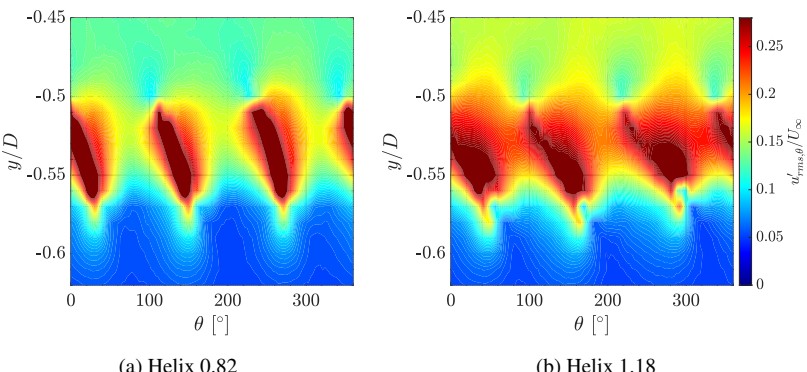

(a) Helix 0.82                                          (b) Helix 1.18

**Figure 15.** Normalized phase-locked RMS-values of the streamwise fluctuations $u'_{rms,\theta}/U_\infty$ at a downstream position of $x/D = 0.35$.

additional measurements are conducted. The spatial distance between the points is fixed to $\Delta(y/D) = \Delta(x/D) = 0.01$ in each direction. The contour plots in figure 15 show the normalized phase-locked RMS-values of the streamwise fluctuations $u'_{rms,\theta}/U_\infty$ over one rotation at the closest distance $x/D = 0.35$ for both actuation cases. As already mentioned before,

both cases experience high fluctuation levels of around $u'_{rms,\theta}/U_\infty \approx 0.25$ between lateral locations $y/D = (-0.5, -0.57)$. However, the Helix 1.18 contours are more smeary, and hence, a more constant input of fluctuation is seen over one rotor revolution. As already mentioned, this representation does not show the realistic vortex shedding since the actuation is out of sync to the rotational frequency, which is the basis for the phase-locking procedure. The phenomena shown here describe multiple flow situations that occur throughout the actuation time.

To get a more thorough view on the whole area of the refined grid, phase-locked contour plots of the entire refined measurement x-y-plane are shown in Figure 16. In this figure, the normalized phase-locked vorticity in z-direction (out of plane) $\xi_z$ is shown for the azimuthal blade position $\theta = 90°$. The figures are screenshots of a video consisting of multiple phase locked positions corresponding to azimuthal blade positions $\theta = 0 - 360°$. The videos for the two Helix cases can be found under the link Youtube.com/blbllablalbla. The normalized vorticity in z-direction is calculated as follows:

$$\xi_z = \frac{D}{U_\infty}\omega_z = \frac{D}{U_\infty}\left(\frac{\partial v}{\partial x} - \frac{\partial u}{\partial y}\right) \tag{3}$$

Once again, both actuated cases are directly compared in the figure. Here, vortex-sheet-shaped elongated structures travel downstream and broaden. Nevertheless, since solely phase-locked results are shown, which were locked with the rotational frequency $f_r$ and not with the additional rotational frequency $f_{r,a}$, the "vortex sheets" visualize an averaged representation of the flow structures travelling downstream. For the Helix 0.82 case, the two vortices at the extreme $y/D$-positions move

downstream with an almost identically speed. A C-shaped lateral elongated structure corresponding to the sinusoidal vortex shedding with different vortex strengths forms. In contrast to that, the Helix 1.18 case shows an additional delay in the shedding of the two extreme positions. A diagonal-shaped structure travels downstream in the phase-locked flow field.

To understand the elongated vortex structures in Figure 16 better, two additional constant-pitch experiments are conducted. Within these measurements, the additional blade pitch is changed to a constant value matching either the minimum or the

WIND
ENERGY
SCIENCE
DISCUSSIONS

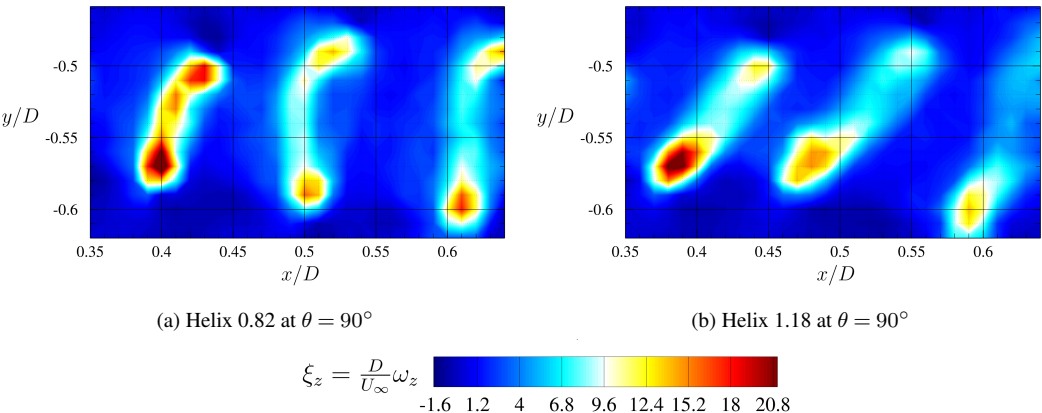

(a) Helix 0.82 at $\theta = 90°$          (b) Helix 1.18 at $\theta = 90°$

$$\xi_z = \frac{D}{U_\infty}\omega_z$$

-1.6  1.2  4  6.8  9.6  12.4  15.2  18  20.8

**Figure 16.** Normalized phase-locked vorticity in z-direction $\xi_z$ at azimuthal position $\theta = 90°$ for a) Helix 0.82 and b) Helix 1.18. Video link: https://youtu.be/ta3KwE5yuSQ.

maximum of the dynamic blade pitch motion of the actuated cases $\Delta\beta = const. = \pm4°$. Figure 17 shows the vortex shedding (visualized by $u'_{rms,\theta}/U_\infty$) over one rotor revolution for the extreme fixed pitch cases together with a sketch of the stream tube. For the $\Delta\beta = -4°$ case, blade loading and consequently $C_T$ is increasing. Thus, a stronger vortex forms at the blade tip. With an increased $C_T$, the stream tube is expanding leading to the tip vortex being shed rather outside at around $y/D = -0.57$. The opposite happens for the $\Delta\beta = +4°$ case, where the high pitch angle leads to an unloaded turbine blade, with a decreased

$C_T$. Consequently, weaker tip vortices form and the wake stream tube is not widening up as for lower blade pitch angles and travels downstream at a more inward position of around $y/D = -0.50$. The lateral position of the detectable vortices for the fixed-pitch cases match the maximum values measured for the dynamically pitched Helix cases in figure 15 and explain the elongated shape of the tip vortices.

    To get the full view of the transient flow behavior in the wake of the dynamically actuated wind turbine, further analysis can

be conducted. Instead of phase-locking the data with the rotor azimuth as presented in this chapter, the additional excitation frequency $f_e$ has to be determined and used for phase locking to see and understand what is happening with the wake flow during one additional rotation which is created by the Helix control.

### 4.2.3 Wake flow analysis: results phase-locked with the additional frequency

    In the following, phase-locked results are presented which use the additional excitation frequency $f_e = 2.5\,Hz$ as the phase-

locking frequency. Since the measured data solely consist of the azimuthal and the blade 1 pitch positions, the additional frequency signal has to be extracted from these signal at first. This is done by taking the envelope/beat frequency of the superposition of the two acquired signals. Output of this pre-processing step is a sinusoidal signal with $f_e = 2.5\,Hz$ that is used in the following to phase-lock the measured FRAP data.



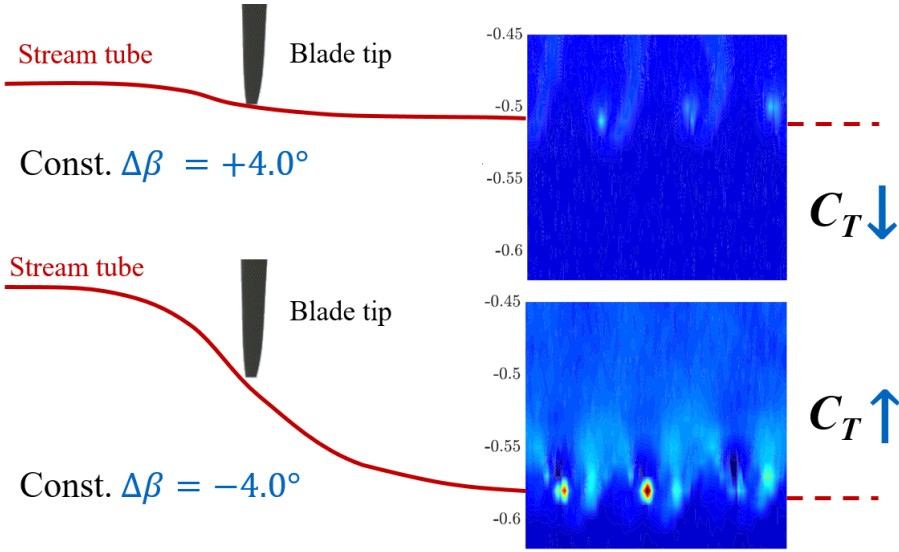

**Figure 17.** Normalized phase-locked RMS-values of the streamwise fluctuations $u'_{rms,\theta}/U_\infty$ for fixed pitch $\Delta\beta = -4°$ and $\Delta\beta = 4°$ at a downstream position of $x/D = 0.35$.

In Figure 18, the results of the phase-locking process with the additional frequency are compared for both actuated cases
(Helix 0.82 top and Helix 1.18 bottom). The contour plots show the normalized phase-locked vorticity in z-direction (out of plane) $\xi_z$, which is calculated according to Equation 3. In this way, the results can be compared to the ones from Figure 16, which are representative for the phase-locking analysis with the azimuthal position . The figures show snapshots of multiple phase-locked positions for one complete phase-locking period $T_{pl}$. The six contour plots are taken with time steps of $\Delta t = 1/6 \cdot T_{pl}$. The videos for the two Helix cases can be found under the link Youtube.com/link.
When analysing the results in Figure 18, it is clearly visible that different flow structures can be observed dependent on the locking frequency. Due to the rather long phase-locking period applied in this section ($T_{pl} = 1/f_e$), the contours do not show the typical shedding of the three vortices, but rather a vortex path, which shows the shedding locations of multiple blade tip vortices over time. During one revolution of the additional rotational component, the tip vortex at $x/D = 0.35$ is starting at $y/D \approx -0.50$ at $t = 1/6 \cdot T_{pl}$ and travelling laterally outwards until $y/D \approx -0.57$ at $t = 3/6 \cdot T_{pl}$, from where it is moving back
towards the wake center. Whereas, at this movement, a weakening of the magnitude can be observed. Looking at the farthest downstream location of the measurement plane at $x/D = 0.64$, the vortex path is slightly lacking behind in time and moreover shows an even wider lateral expansion as at $x/D = 0.35$. Hence, this indicates an increasing mixing effect at more downstream positions. The observations taken from the phase-locked data show that the tip vortex path performs a meandering movement, while, the tip vortices for the two Helix cases look quite similar, both in shape and in magnitude. These observations confirm
the conclusions drawn from the vorticity magnitudes in the results phase-locked with the rotor azimuthal position in Figure 16. These are backed by the findings of the $C_T$ graph and the lateral tip vortex position changes for the constant extreme pitch angles $\Delta\beta = const. = \pm 4°$. Due to the changing blade pitch $\beta$ and thus a varying $C_T$, the vortex shedding is travelling back





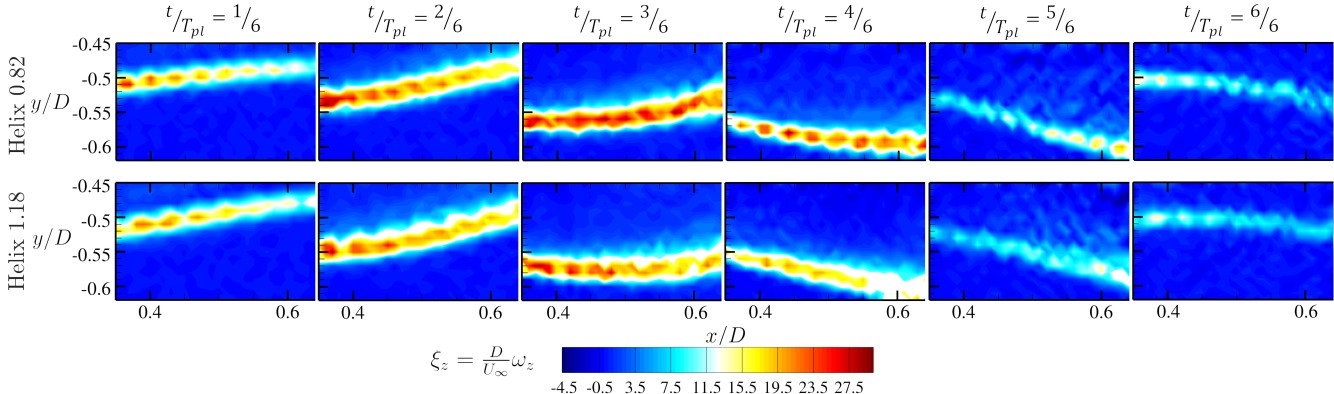

**Figure 18.** Normalized phase-locked vorticity in z-direction $\xi_z$ taken from the 12 s video for one additional rotation at the time steps $t/T_{pl} = 1/6$, $t/T_{pl} = 2/6$, $t/T_{pl} = 3/6$, $t/T_{pl} = 4/6$, $t/T_{pl} = 5/6$ and $t/T_{pl} = 6/6$, for Helix 0.82 (top) and Helix 1.18 (bottom). Video link: https://youtu.be/0x432E6h-7E.

and forth in lateral direction over time. Furthermore, the results confirm that a traditional vortex interaction mechanism through leapfrogging as seen in the reference case is not present in the Helix actuation. Consequently, the meandering of the vortex in the blade tip region is the main driver for the increased wake mixing for the Helix technique.

### 4.2.4 Out-of-plane velocity

This last results section is intended to explain the differences between the Helix 0.82 and Helix 1.18 cases. For this, an analysis of the rotational velocity component is conducted. In the presented case, in which a x-y plane is measured, the velocity component in the out-of-plane, z-direction $w$ is representing the rotational direction of the wake. As already mentioned in the explanation of the control technique in section 2, the actuation of the out-of-sync blade pitch provokes a clockwise or counter clockwise additional rotation of the wake. Figure 19 shows contours of the investigated flow field of the normalized averaged velocity in the out-of-plane component $\bar{w}/U_\infty$. It can be seen that the two Helix cases $\bar{w}/U_\infty$ are in significant contrast to each other. For the Helix 0.82 case, the velocity $\bar{w}/U_\infty$ in the inner part of the wake $|y/D| < 0.5$ is positive, whereas the outer part $|y/D| > 0.5$ is slightly negative pointing in the same direction as the additional rotation of the 0.82 Helix excitation. For this case the w-velocity magnitude is rather low as the additional rotational component is opposite to the wake rotation. The Helix 1.18 case, in contrast, is showing a contrary behaviour. The inner part shows slightly negative velocities, whereas the outer part is rotating in CCW rotation. Here the additional rotational component of the Helix is adding to the wake rotation. Hence, $\bar{w}/U_\infty$ has a larger magnitude compared to the Helix 0.82 case.

The formation of this contrary sheared flow can be observed when looking into the refined tip vortex region, which is presented in Figure 20 showing contours of the normalized velocity component $w/U_\infty$, phase-locked with the additional frequency taken from a 12 s video for one additional rotation at six time steps from $t/T_{pl} = 1/6$ to $t/T_{pl} = 6/6$. For the Helix 0.82 case presented on top, the meandering shear border is located towards the center of the wake. The phase-locked velocity



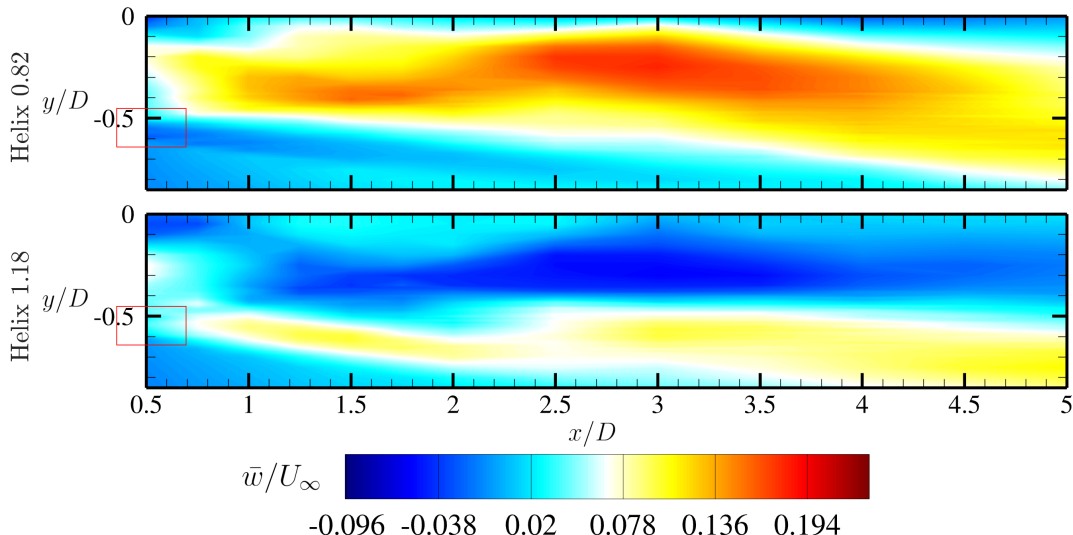

**Figure 19.** Contour of flow field for normalized average velocity in the out-of-plane component $\bar{w}/U_\infty$, for the two actuated cases Helix 0.82 (top) and Helix 1.18 (bottom). The red square defines the plane of the refined tip vortex analysis.

$\bar{w}/U_\infty$ is positive in the inner section at the beginning of the period $T_{pl}$ and is changing its sign when meandering outwards $t/T_{pl} > 2/6$. The opposite can be observed for the Helix 1.18 case, where a change from negative to positive velocities is observed after the wake is meandering from the centre $t/T_{pl} \leq 2/6$ outward $t/T_{pl} \geq 3/6$. The strongest magnitudes can be observed when the meandering border reaches the utmost point at time steps $t/T_{pl} = 3/6$ and $t/T_{pl} = 4/6$. Here the observation of the averaged velocity component is confirmed. For $\bar{w}/U_\infty$ in the Helix 0.82 case, a small velocity magnitude in the direction of wake rotation can be seen. However, for the Helix 1.18 case, the magnitude is significantly bigger, pointing in the same direction as the wake rotation.

Due to the larger rotational velocity observed in Figures 19 and 20, a stronger sheared flow between the inner wake region and the more energetic outer flow is introduced in the Helix 1.18 case. This is the reason for the faster wake recovery observed in the initial turbine setup analysis with two tandem turbines. Even though the analysis demonstrates the differences and a possible explanation between the Helix 0.82 and 1.18 cases from the FRAP measurements, further detailed analysis of the flow is necessary to fully understand the wake aerodynamics and the detailed mechanisms responsible for the different wake recovery for the Helix 0.82 and 1.18 cases.

## 5 Conclusions and outlook

In this article, the potential of the Helix control technique was investigated by experimental wind tunnel tests. In the introduction, a motivation for investigating the relatively new and thus not completely understood Helix approach was provided and the research questions on the Helix approach and the wake mechanism were stated. In the following section, the Helix control

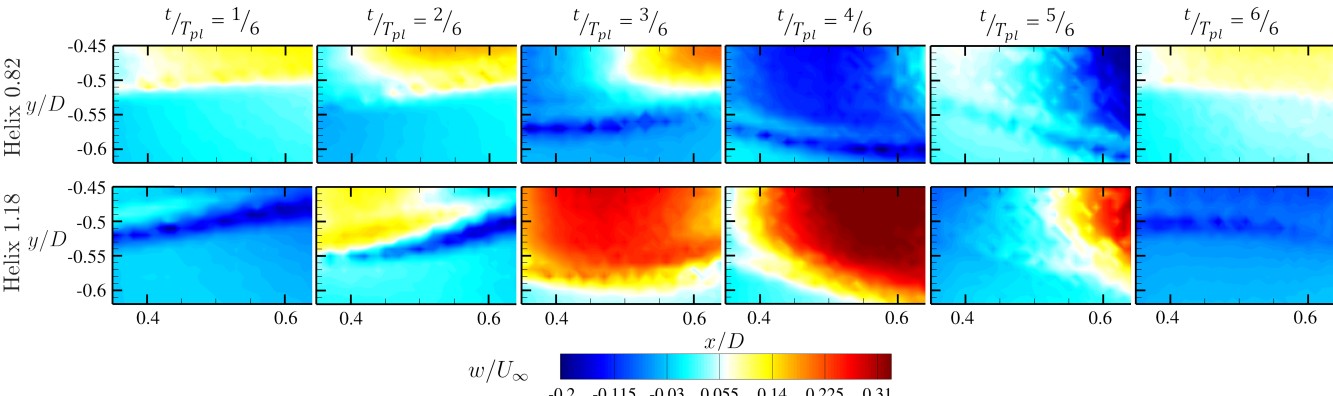

**Figure 20.** Normalized phase-locked velocity component $w/U_\infty$ in z-direction $\xi_z$ taken from the 12 s video for one additional rotation at the time steps $t/T_{pl} = 1/6$, $t/T_{pl} = 2/6$, $t/T_{pl} = 3/6$, $t/T_{pl} = 4/6$, $t/T_{pl} = 5/6$ and $t/T_{pl} = 6/6$, for Helix 0.82 (top) and Helix 1.18 (bottom). Video link: https://youtu.be/KH06Y7n5_gA.

technique was introduced and the pitch excitations causing the additional rotational movement were explained. The experimental setup section presented the G1 model wind turbines and the atmospheric boundary layer wind tunnel at TUM used for the measurements. Furthermore, the different measurement stages with the tandem turbine and single turbine setup were shown. At the end of this section, a detailed description of the FRAP probe, which was used for the flow measurements, was provided. In the results section, the findings of the measurements were presented and discussed. Results of the first measurement stage,
the turbine data, were presented and the influence of applying the Helix technique and dependencies towards the actuation frequency was analysed. For the wake data, the time-averaged wake aerodynamics was presented first. Furthermore, two different phase-locking analyses were discussed: flow measurements phase-locked with the rotor azimuthal position $\theta$ and results phase-locked with the additional frequency $f_e$. At the end of the analysis part, results of the out of plane velocity component $w$ were provided in order to explain the differences between Helix 0.82 and Helix 1.18 actuations.

The presented tests were conducted under laminar inflow conditions to intentionally highlight the flow effects in the wake and not blend the effects due to a rather high however more realistic inflow turbulence intensity. Further, it is noted again, that these conditions are not realistic as it would be seen by a full-scale wind turbine (farm). Consequently, the results should be considered as such and not taken as absolute gains for a real-world application but rather as trends and potentials when applying the Helix control strategy.

The general findings and conclusions of this study are summarized in the following:

- Both Helix cases (CW and CCW rotation) are characterized by a significantly faster wake recovery compared to the baseline case, which can be seen in the high energetic content in the wake experienced by the sensor turbine.

- The performance gains occur at similar additional Strouhal numbers $St_{add}$ with a distinct maxima for both actuated cases.



• If the helix control is applied, the loads of the actuated turbine and the sensor turbine operating in the wake are increased compared to the baseline case, whereas this accretion is more dominant for the actuated turbine.

• The wake has higher turbulent fluctuation, stronger interaction with the outer flow and faster wake expansion whenever the Helix actuation is applied. A detailed view into the tip vortex shedding indicates that no conventional vortex interaction mechanism like leapfrogging occurs in the wake of the actuated turbine.

• The higher turbulence intensity levels and the increased mixing are introduced from a radial tip vortex meandering, which is causing the vortices to interact faster, shortening the shielding mechanism of the vortices and thus leading to faster wake recovery.

• The Helix 1.18 case with CCW rotation has slightly higher gains and a faster wake recovery compared to the Helix 0.82 case with CW, which are also accompanied by higher loads seen by a downstream turbine. These differences are
mainly due to the different rotational direction of the Helix additional excitation, which was mainly demonstrated in the $w$ velocity component in the presented case. This additional component is either acting in the same (Helix 0.82) or in the opposite (Helix 1.18) direction of the wake rotation and thus increasing mixing for the Helix 1.18 case compared to the Helix 0.82 one.

• The findings suggest to prefer the slower actuation $f_\beta/f_r < 1.0$ over $f_\beta/f_r > 1.0$. Even though the wake mixing is
stronger for Helix 1.18, the power gains of the two different actuation settings are almost identical. However, when also including the loads and duty cycles, the Helix 0.82 case it to be favored.

Hence, the results collected in this work indicate that the Helix method is a very interesting wake mixing technique and it is a potential alternative or addition to standard wake control.

Nevertheless, the Helix technique and its real-world realization is just at the beginning of its development. Consequently,
this study can only provide a first insight on the potential and wake mixing mechanisms. An investigation of the influence of inflow turbulence on the wake mixing potential could be the next step. Further, testing the Helix technique in wind farm control studies could be promising.

*Video supplement.* The videos for Figures 16, 18 and 20 can be accessed via the following links:

Figure 16: https://youtu.be/ta3KwE5yuSQ
Figure 18: https://youtu.be/0x432E6h-7E
Figure 20: https://youtu.be/KH06Y7n5_gA

*Author contributions.* Franz V. Mühle and Florian M. Heckmeier contributed equally to this work. They conducted the experiments, evaluated the data and wrote the article. Filippo Campagnolo helped with the wind tunnel experiments. Furthermore, he evaluated and analyzed the turbine loads and wrote the article chapter. Christian Breitsamter supervised the work and proof read and corrected the article manuscript.



*Competing interests.*    The authors declare that they have no conflict of interest.





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
