# Peer review of "Wind tunnel investigations of an individual pitch control strategy for wind farm power optimization"

_Wind Energy Science, 2023_

## Author Comment (AC1)

**Authors' response to Referee 2**

**General**

**This article presents a wind tunnel study of a wake control strategy named 'Helix'. The article is of relevance to wind energy community and fits within the scope of the journal. The study is performed systematically and data quality is good. There are, however, some concerns regarding the work which need to be addressed properly. These are listed below:**

We thank the referee for reviewing this manuscript, the valuable feedback, and the constructive comments. At this stage of the review process, we respond to referee #2's comments and propose improvements for the journal manuscript. The referee's original comments are printed in bold followed by the corresponding answers. A screenshot of the different versions of the updated passages from the manuscript is provided below the answer.

**Specific comments**

1) **My main concern is regarding the blockage effect in the experiments. As authors indicate, the blockage is about 20%, which is considerably high. To tackle this, they propose a 'blockage-corrected free-stream velocity'. Does correcting the free-stream velocity resolve completely the effect of blockage? In principle, your turbine is placed in a confined channel, where the flow acceleration can affect the turbine power output and also affect the development of the wake due to an effective favorable pressure gradient in the flow. How is this addressed in the work? At least, the authors should mention the limitations introduced in the work due to the blockage effect to properly guide the reader.**

Thank you for this important comment. The same concern was also raised by Reviewer #1 and Reviewer #3. We agree that the blockage is very high and was not discussed in an appropriate way. We added information about the blockage effect in section 3.2. We included a paragraph in which we use several studies investigating the blockage effect to discuss the effect that blockage is expected to have on wake development.

however accelerated, resulting in a higher velocity experienced by the turbines.  To account for such an
effect on the turbine performance, it can be corrected by applying analytical models, a recent review is presented in the study by
Steiros et al. (2022). More information is presented by Ross and Polagye (2020) who conducted an experimental assessment
of such models and their application to different wind turbine concepts. In the present study, the performance was corrected
by a calculation of the Rotor Effective Wind Speed (REWS) for the upstream G1, done as described in Campagnolo et al.
(2022). This revealed a blockage-corrected free-stream velocity $U_{\infty,corr} \approx 5.9\ m/s$, which correlates to the rated wind speed
of the model wind turbine. Consequently, the turbine is operated at a tip-speed ratio of approx. $\lambda = 8.2$.  Based on the
authors' knowledge, there are no analytical blockage correction models that would allow us to quickly assess the influence
of the wind tunnel walls on the wake. One possible way to investigate this is to use computational fluid dynamics (CFD).
Zaghi et al. (2016) studied the effect of blockage on a model wind turbine with a Reynolds Averaged Navier-Stokes (RANS)
simulation. They analyzed the streamwise wake velocity and found increased velocities in the area behind the rotor but also
in the outer region of the wake in case a wind tunnel wall was present. In a more detailed analysis Sarlak et al. (2016) used
Large Eddy Simulation (LES) combined with the actuator line technique to investigate wake velocity and Reynolds stresses
for different blockage ratios up to $\alpha = 0.2$. They found a significant impact on the mean wake velocity in the case of the
highest blockage ratio. Especially in the region outside of the rotor, the velocity is found to increase; this augmentation is
mitigated in the rotor area but is still present. Furthermore, they concluded that blockage has no considerable effect on the
wake mixing rate as maximum and minimum velocities do not differ significantly. In a combined experimental and numerical
study McTavish et al. (2014) investigated the influence of wind tunnel blockage on the wake width and found that the wake
compresses when blockage increases. Consequently, the wake results of the presented study are expected to be characterized
by slightly higher streamwise velocities and a narrower wake compared to a full-scale test. As a result, in the analysis of the

**2) The experiments are performed in an almost laminar uniform flow (Tu<0.5%). I understand that the authors intend to isolate the effect of control strategy. However, the relevancy of the control approach to field conditions with turbulence intensity greater than 5% and boundary layer shear must be discussed. In other words, does the control strategy remain effective at high turbulence intensities and in the presence of flow shear?**

Thank you for raising this point. The same topic was also brought up by referee #3. We agree that it is important to address this point better. We added a small discussion about this in the literature review in the introduction chapter. We added a source of a study where the authors investigate the effect of inflow turbulence on the efficiency of dynamic wake mixing and show that inflow turbulence has a significant inflow on the effectiveness of wake mixing for power optimization. Furthermore, we updated the future works slightly to say that further investigations on inflow turbulence are needed.

approach experimentally in a wind tunnel (W/T) and should give a detailed insight in the wake aerodynamics. To provide
such a detailed insight, the flow in the wind tunnel has to be a clean lab flow, which is uniform and is characterized by a
very small turbulence intensity. Such clean inflow will not only highlight the effects of the control technique in the wake
but also influence their effectiveness. Wake mixing techniques like Helix add turbulence to the wake. Consequently, if the
turbine inflow is already characterized by higher ambient turbulence the effect of wake mixing will be mitigated. In a recent
study Mühle et al. (2024) compare power gains for wake mixing by dynamic yaw for different inflow turbulence. They found
a strong reduction of the effect on the power of a two-turbine setup in case of high inflow turbulence and thus confirm the
findings of Munters and Meyers (2018a). Nevertheless, they suggest that wake mixing has the potential to improve the power
output of a wind farm in case the wakes are strong and persistent.

Nevertheless, the Helix technique and its real-world realization  are just at the beginning of its development. Consequently,
this study can only provide a first insight  into the potential and wake mixing mechanisms.  Additional wake analysis with
different Strouhal numbers is required to prove that the wake meandering is the main driver for increased mixing. Moreover, an
investigation of the influence of inflow turbulence on  wake mixing is needed
to understand its potential in more realistic inflow conditions characterized by higher levels of ambient turbulence. Further,
testing the Helix technique in wind farm control studies could be promising.

**3)** **The authors indicate that the turbine rotation is fixed at an optimum value. How is this optimum value obtained and does it remain the same for the uncontrolled and controlled turbine configurations?**

Thank you for pointing this out. It is obtained to maintain the operational tip-speed ratio of the G1 model turbine λ = 8.2. With the corrected inflow velocity of $U_\infty \approx 5.91$ m/s this results in a rotational velocity for the turbine of 840 rpm. This rotational velocity was also maintained when the Helix control operated the turbine. We added a clarifying sentence in section 3.2.

> varied within the range $10.1 : 0.3 : 17.9\ Hz$.  In the controlled and uncontrolled configuration, the upstream turbine is op-
> erated with a constant rotational frequency of $f_r = 840\ rpm/60 = 14\ Hz$ and an optimal collective pitch offset of $\beta_0 = 0.4°$.
> 220   With these adjustments, the turbine operates at the desired G1 tip-speed ratio $\lambda = 8.2$ at $U_{\infty,corr} \approx 5.9\ m/s$. These controls

**4)** **The pressure probe measurements are performed for 40 seconds. Is that time interval sufficient to give converged flow statistics?**

Thank you for this remark. We have carried out preliminary tests in this regard, which have shown that the relevant content in terms of flow statistics is recorded with a measurement time of 40 seconds. For example, with the additional frequency of 2.5Hz introduced by the helix control, 100 such events occur. We added a clarifying sentence in section 3.2.

> within the range $0 : 0.052 : 0.73$ both in CW and CCW direction. For each investigated actuation frequency, the measurement
> time is set to $t_s = 40.0\ s$. With this recording time and the beat frequency of $f_e = 2.5\ Hz$, the rotation of the fixed-frame
> moments induced by the Helix control is recorded 100 times in one measurement interval. The downstream turbine (sensor

**5)** **The baseline plots in figure 7 are very hard to distinguish from the background of the plot. Consider improving the figure.**

Thank you for the hint. We agree that the lines are, at first sight, hard to detect. However, we decided on purpose to have these lines not as prominent as the ones for the actuated cases, as they only represent the baseline cases and are constant anyway. The main message of these plots is provided by the thick lines depicting the data of the actuated cases, and we did not want to draw attention away from those. When the reader first sees the graph, he immediately notices them and thus the two peaks for CW and CCW rotation; when focusing on the graph a bit longer, he also sees the secondary information, which is the reference cases. Consequently, we decided to leave them as they are.

**6)** **The authors compare the trend in the thrust coefficient with that in the available power, and identify some differences. Is that a fair comparison? If so, what is the possible explanation for the difference? Wouldn't it be more appropriate to compare thrust coefficient with power coefficient?**

Thank you for this comment. We agree with the referee that looking at the thrust coefficient and normalized power is unfair. We changed the figure, which now shows the thrust normalized with the thrust measured for the baseline case. The thrust coefficient was calculated using an estimate of the rotor effective wind speed, which can be affected by uncertainty when the Helix is active. We added some discussion about the observed trend for power and thrust, highlighting that additional analysis are needed.

In general, the extracted power of the actuated turbine decreases for all  frequencies, since it is operated in a non-optimal operating point. With an increasing actuation frequency, this effect is also increasing. The decrease is particularly significant (around 30%) for the highest actuation frequencies, and is remarkably higher than the values (few percent) noted in previous research works based on CFD (Frederik et al., 2020a) and aeroelastic (Taschner et al., 2023b) simulations. Whatever the reason
355      for this difference may be – different performance of the G1 compared to that of a full-scale machine, wall blockage affected by the actuation frequency, physical effects not modeled in the simulation environments – further investigation is needed.

370      Figure 7 shows the thrust  $T$ of the actuated upstream turbine  normalized by the thrust of the upstream/actuated turbine in the baseline scenario $T^*$. The trend for  $T$ differs from the one of the  normalized power of the upstream turbine in Figure 6: the thrust is indeed reduced by the Helix but remains quite constant with increasing actuation frequencies
375       (only a minor reduction is observed). Once again, the reason for the different trends observed for power and thrust requires further analysis.

**7) Is the blade pitch synchronized with the rotor rotation for all the cases?**

Thank you for the question. No, it is not. We added a paragraph in section 3.3 that comments on this aspect and its impact on the results, which is not present.

230       During each experiment, the Helix was activated a few seconds before the start of the acquisition, to allow for the downstream propagation of the new wake. The activation of the Helix, however, was not synchronized to a specific azimuth position of the rotor. For $f_\beta \neq f_r$ this aspect is not relevant and does not affect the results. In this regard, each plot in Fig. 3 shows, for 5 of the conducted experiments, the distribution of the first blade azimuth ($\theta_1$) detected when the required pitch is within the range $0.98\beta_{max} < \beta_1 < 1.02\beta_{max}$, with $\beta_{max} = \hat{\beta} + \beta_0 = 4° + 0.4°$ the maximum requested pitch
235      angle. The dashed black line, instead, marks the azimuth position ($\theta_{1,ini}$) at the very first time the required pitch is within the range $0.98\beta_{max} < \beta_1 < 1.02\beta_{max}$. For experiments conducted with $f_\beta \neq f_r$, the pitch is detected around $\beta_{max}$ for the whole range of azimuth positions (the distribution is not completely homogeneous solely due to the discrete sampling), regardless of the value of $\theta_{1,ini}$. This implies that the fixed-frame moments produced by the Helix rotate over the entire range of azimuth positions, which leads to the expected effects on the wake.
240      Different is the case with $f_\beta = f_r$; the maximum pitch is indeed always detected at the same azimuth position, i.e. around $\theta_{ini}$. Previous experimental (Campagnolo et al., 2016) and numerical (Fleming et al., 2014) investigations found that the azimuth position of maximum pitch affects the achieved amount of wake deflection. However, the study of Wang et al. (2016) revealed that the power gains observed on the downstream machine may only be a little higher than the power losses experienced by the upstream machine, at the price of a significantly increased loading of the upstream machine. In this article, it was therefore
245      preferred to exclude the case $f_\beta = f_r$ from the following discussions. A complete analysis would, indeed, require investigating

9

[Figure]

**Figure 3.** Distributions of the first blade azimuth ($\theta_1$) detected when the required pitch is close to $\beta_{max}$, and with $f_\beta/f_r = [0.72, 0.86, 1.12, 1.28, 1]$. The dashed black lines mark $\theta_{1,ini}$, i.e. the azimuth position recorded at the very first time the required pitch is within the range $0.98\beta_{max} < \beta_1 < 1.02\beta_{max}$.

the effect of the azimuth position of maximum pitch. Furthermore, previous results have shown the poor performance of this specific implementation of the Helix, thus making it uninteresting.

**8) For phase-locked measurements, how is it ensured that during a certain azimuthal phase the pitching phase is also the same for all the cases?**

Thank you for this comment. We try to clarify our approach in the following. The situation that you describe wasn't ensured during our study. We would have needed to measure the flow within the wake for several minutes, to have a sufficient amount of such events, and thus multiple exact matchings. We see the effect of this shortcoming in the Figure below, and the corresponding discussion in the text.

[Figure]

(a) Helix 0.82 at $\theta = 90°$        (b) Helix 1.18 at $\theta = 90°$

$$\xi_z = \frac{D}{U_\infty}\omega_z$$

-1.6   1.2    4    6.8   9.6   12.4   15.2   18   20.8

**Figure 15.** Normalized phase-locked vorticity in z-direction $\xi_z$ at azimuthal position $\theta = 90°$ for a) Helix 0.82 and b) Helix 1.18. Video link: https://youtu.be/ta3KwE5yuSQ.

By applying the additional phase locking with the beat frequency, we tried to overcome this shortcoming. We added some further explanation in section 4.2.3, on how we extracted the envelope/beat frequency data (see Reviewer #1 Question 12)

a sinusoidal signal with $f_e = 2.5\ Hz$ that is used in the following to phase-lock the measured FRAP data. The process of extracting the envelope/beat frequency from the blade azimuth and pitch position is visualized in Figure 17. After extracting the time series for the beat frequency for each individual measurement point, the measured FRAP data are phase-averaged with the beat frequency. Readings of all measurement points can thereby be correlated.

[Figure]

**Figure 17.** Schematic visualization of the process of extracting the envelope/beat frequency from the blade azimuthal and pitch position for phase-locking with the additional frequency.

**Technical comments**

**There are several minor grammatical mistakes throughout the article, which need to be addressed:**
**Line 308: as seen in figure 8**

Thank you for pointing out this typo. We changed that.

**line 31 ('turbine excitation "triggers" wake meandering')**

Thank you for this remark. We changed it accordingly.

**line 101 ('dynamic variation')**

Thank you. We corrected it.

**line 293 ('does not apply to')**

Thank you for this remark. We changed it.

**line 308 ('as seen in')**

Thank you for pointing out this typo. We changed that.

**line 396 (is 'data basis' a correct word?)**

Thank you. We removed "data" from the sentence.

**line 411 (sounds a bit repetitive)**

Thank you for pointing this out. This is a mistake and was also clarified in the comments of Reviewer 1. We corrected the second sentence with "x/D=5.0".

**line 439 (the link is missing)**

Thank you for pointing this out; we added the final link.

---

## Author Comment (AC2)

**Authors' response to Referee 1**

**General**

**The paper „Wind tunnel investigations of an individual pitch control strategy for wind farm power optimization" investigates the impact of the Helix approach for wake control on the total power output of two turbines in tandem configuration as well as the effect of the Helix control on the wake development. The paper is well written and the analysis presented gives good insight in what is happening in the manipulated wake. The procedure of the Helix control and the contribution of the respective Nevertheless, there are a few points the authors should address and clarify, respectively.**

We thank the referee for reviewing this manuscript, the valuable feedback, and the constructive comments. At this stage of the review process, we respond to referee #1's comments and propose improvements for the journal manuscript. The referee's original comments are printed in bold followed by the corresponding answers. A screenshot of the different versions of the updated passages from the manuscript is provided below the answer.

**Specific comments**

1) **The authors mention, that the blockage in this experiment is rather high and present a correction for the inflow velocity. Since that correction is needed, can the authors also say anything about the possible impact of the blockage on the results of the measured wake? It would be helpful to get a feeling if these results are representative for the Helix control or if it might be an artefact. Are there aspects from other investigations with lower blockage that are comparable?**

Thank you for this important comment. The same concern was also raised by Reviewer #2 and Reviewer #3. We agree that the blockage is very high and was not discussed appropriately. We added information about the blockage effect in section 3.2. We included a paragraph in which we present several studies investigating the blockage effect to discuss the effect that blockage is expected to have on wake development.

however accelerated, resulting in a higher velocity experienced by the turbines.  To account for such an effect on the turbine performance, it can be corrected by applying analytical models, a recent review is presented in the study by Steiros et al. (2022). More information is presented by Ross and Polagye (2020) who conducted an experimental assessment of such models and their application to different wind turbine concepts. In the present study, the performance was corrected by a calculation of the Rotor Effective Wind Speed (REWS) for the upstream G1, done as described in Campagnolo et al. (2022). This revealed a blockage-corrected free-stream velocity $U_{\infty,corr} \approx 5.9\ m/s$, which correlates to the rated wind speed of the model wind turbine. Consequently, the turbine is operated at a tip-speed ratio of approx. $\lambda = 8.2$.  Based on the authors' knowledge, there are no analytical blockage correction models that would allow us to quickly assess the influence of the wind tunnel walls on the wake. One possible way to investigate this is to use computational fluid dynamics (CFD). Zaghi et al. (2016) studied the effect of blockage on a model wind turbine with a Reynolds Averaged Navier-Stokes (RANS) simulation. They analyzed the streamwise wake velocity and found increased velocities in the area behind the rotor but also in the outer region of the wake in case a wind tunnel wall was present. In a more detailed analysis Sarlak et al. (2016) used Large Eddy Simulation (LES) combined with the actuator line technique to investigate wake velocity and Reynolds stresses for different blockage ratios up to $\alpha = 0.2$. They found a significant impact on the mean wake velocity in the case of the highest blockage ratio. Especially in the region outside of the rotor, the velocity is found to increase; this augmentation is mitigated in the rotor area but is still present. Furthermore, they concluded that blockage has no considerable effect on the wake mixing rate as maximum and minimum velocities do not differ significantly. In a combined experimental and numerical study McTavish et al. (2014) investigated the influence of wind tunnel blockage on the wake width and found that the wake compresses when blockage increases. Consequently, the wake results of the presented study are expected to be characterized by slightly higher streamwise velocities and a narrower wake compared to a full-scale test. As a result, in the analysis of the

2) **The authors mention that they expect an impact of higher turbulence intensity on the results with which I totally agree. What about different wind velocities? I assume that this control is applied in the partial load region. Do the authors expect comparable results for the complete wind velocity range in the partial load region?**

Thank you for pointing this out. We added a paragraph within the conclusions that discusses the expected behavior of Helix for the complete wind velocity range in the partial load region.

The tests were conducted at lower than-rated wind speeds (partial load region, or Region II), in which, usually, a wind turbine is operated with a constant blade pitch and tip speed ratio. Neglecting spatial and turbulent-induced variations of the wind speed, the resulting distribution of the angle of attack along the blade span does not depend on the mean wind speed. It follows that the distributions of the axial induction coefficient and non-dimensional circulation along the blade span, the intensity of the trailed vorticity shed by the blades, and the pitch of the helical vortex, do not vary with the mean wind speed.

29

Although not supported by experimental evidence, it is therefore reasonable to assume that the results, here gathered at a lower-than-rated wind speed, can be similarly observed over the entire range of Region II wind speeds.

3) **When looking at the total power of the two turbines the second turbine is also running at a constant rotational velocity. The second turbine clearly sees different and non-uniform and temporally changing inflow condition to which an activated turbine control would react to. Did the authors try to activate the „normal" control of their turbine? What effect did that have on the total power? Why did they decide to run the second turbine also at a constant rotational frequency?**

Thank you for pointing this out. We agree that the sensor turbine sees varying inflow conditions. We did not activate the "normal" control of the downstream turbine because we decided to use it purely as a sensor turbine and did not want to change anything in the turbine's operation so that it has the same conditions for all investigated cases. To account for the lower wake velocity, we performed velocity measurements at the location of the downstream turbine before the tests of the tandem setup with the upstream turbine not controlled. This was then used to adjust the operational settings of the downstream turbine. We did not test it with the "normal" control active, but we think the effect on the total power would only be minor, as the $C_P$-Lambda curve of the G1 is relatively flat around TSR=8.2. We added an explanation for this in section 3.3.

moments induced by the Helix control is recorded 100 times in one measurement interval. The downstream turbine (sensor G1), instead, serves as a sensor. To this aim, it is down-rated ~~to $f_r = 750~rpm$ and has a~~ and operates at a constant rotational velocity of $f_r = 750~rpm$, with a pitch offset of $\beta_0 = 0°$. This rotational velocity was adjusted based on wake measurements at the location of the downstream turbine behind the uncontrolled upstream turbine so that the tip speed ratio of the sensor turbine was approx. $\lambda = 8.2$. Around this tip speed ratio, the $C_P - \lambda$ curve of the G1 is rather flat. Changing inflow conditions due to the Helix should therefore only have a minor impact on the power coefficient of the sensor turbine.

4) **The traversing system and the support of the five hole probe looks pretty massive in figure 2 and can affect the measurements. Can the authors give more details about the distance of the sensing head to the support? Are they sure that the measurements are not influenced by the traverse?**

Thank you for this hint. We carefully studied the design of the wind tunnel traversing system and the probe mount. In the last 15 years, multiple studies conducted with this traversing system have been published (mostly by researchers at the Chair of Aerodynamics and Fluid Mechanics at TUM). Furthermore, more recent studies show a comparison of the FRAP/5-hole probe and a triple-wire hot wire probe with CFD results (e.g. studies by Ruhland et al. (PDF) TRANSPORT AIRCRAFT WING INVESTIGATIONS AIMED ON WAKE VORTEX IMPACT BY OSCILLATING FLAPS (researchgate.net) and Experimental and numerical analysis of wake vortex evolution behind transport aircraft with oscillating flaps | Request PDF (researchgate.net)). Here, we could show that the disturbances at the probe tip are minimal.

5) **Section 3.4 gives many details on the five hole probe and the calibration. Since these details can be found in other also mentioned publications, I recommend to leave that section out of the paper since it is not contributing to the overall story.**

Thank you for this comment. We removed this section, added the following lines in the "Measurement stages" section, and cited the respective research.

The spatial and temporal characteristics and the underlying calibration process of the FRAP can be found in the literature (see Heckmeier et al. (2019); Heckmeier and Breitsamter (2020); Heckmeier et al. (2021); Heckmeier (2022)): A high spatial accuracy below $0.2°$ in both flow angles and $0.1\ m/s$ in the reconstructed velocity can be achieved. The spatial and temporal

[Figure]

**Figure 4.** Schematic fast-response five-hole probe and G1 wind turbine model measurement setup for TUM-AER W/T-C

resolution of the applied FRAP has been investigated and hence, shows the suitability of the usage of the FRAP for this experiment.

**6) In figure 8, the authors observe a ditch in the thrust coefficient for St_add = 0. Do they have any idea where this is coming from?**

Thank you for this question. The case with St_add = 0 has been already studied in previous publications (Wang, Bottasso, & Campagnolo, 2017) (Fleming, et al., 2014). We added a paragraph in section 3.3 that quickly summarizes the results found therein, which showed poor performance of this specific implementation of the Helix, as well as its dependency on the azimuthal position of maximum pitch (something we did not investigate in our work). For these reasons, we decided not to include, in the rest of the paper, the data we gathered with St_add=0.

240     Different is the case with $f_\beta = f_r$: the maximum pitch is indeed always detected at the same azimuth position, i.e. around $\theta_{ini}$. Previous experimental (Campagnolo et al., 2016) and numerical (Fleming et al., 2014) investigations found that the azimuth position of maximum pitch affects the achieved amount of wake deflection. However, the study of Wang et al. (2016) revealed that the power gains observed on the downstream machine may only be a little higher than the power losses experienced by the upstream machine, at the price of a significantly increased loading of the upstream machine. In this article, it was therefore

245  preferred to exclude the case $f_\beta = f_r$ from the following discussions. A complete analysis would, indeed, require investigating

[Figure]

**Figure 3.** Distributions of the first blade azimuth ($\theta_1$) detected when the required pitch is close to $\beta_{max}$, and with $f_\beta/f_r = [0.72, 0.86, 1.12, 1.28, 1]$. The dashed black lines mark $\theta_{1,ini}$, i.e. the azimuth position recorded at the very first time the required pitch is within the range $0.98\beta_{max} < \beta_1 < 1.02\beta_{max}$.

the effect of the azimuth position of maximum pitch. Furthermore, previous results have shown the poor performance of this specific implementation of the Helix, thus making it uninteresting.

**7) In figure 9 the authors leave out the results for St_add = 0. They mention, that it more or less matched the results from another investigation but without load feedback. I don't understand why this one point right in the middle of the curve should be left out. I would highly recommend to add that point also to provide the complete picture.**

Thank you for this question. See reply to point 6.

**8) The presented DELs show a clear increase especially for the first turbine. Unfortunately I have not knowledge or feeling what that would mean e.g. for a real turbine. Even though the authors state in the end, that the results can not directly be applied to real turbines they should at least discuss what an increase of the DELs would have for consequences for real turbines. Would that be a total show stopper for already existing machines or would that addition be within a range that is accounted for in the current design process?**

The design of most wind turbine components, such as the tower, blades, or main shaft, is mainly driven by: 1) the need to withstand the expected fatigue loads over their entire lifespan; 2) the need to withstand the ultimate loads that might act on the wind turbine even only once in its lifetime. To which extent the design is driven by the fatigue or ultimate loads, depends on the machine itself and the component under consideration. The impact of the increased fatigue loads induced by the Helix should, therefore, be assessed specifically for each machine on which it is to be implemented, and can be more or less significant depending on the role played by fatigue in the design process. We added this paragraph to the section 4.1 of the paper.

The design of most wind turbine components, such as the tower, blades, or main shaft, is mainly driven by: 1) the need to withstand the expected fatigue loads over their entire lifespan; 2) the need to withstand the ultimate loads that might act on the wind turbine even only once in its lifetime. To which extent the design is driven by the fatigue or ultimate loads, depends on the

16

[Figure]

Figure 8. Normalized DELs of the upstream/actuated turbine and downstream/sensor turbine for a) rotating hub and b) fixed hub, for changing pitch frequencies $f_\beta/f_r = (0.72 : 0.02 : 1.28)$ compared to the baseline case without any actuation $f_\beta = 0$ (dashed line).

machine itself and the component under consideration. The impact of the increased fatigue loads induced by the Helix should,
410 therefore, be assessed specifically for each machine on which it is to be implemented, and can be more or less significant depending on the role played by fatigue in the design process.

**9) In the text (page 17, lower part) the authors mention four points P1 to P4 for which the spectra are presented in figure 12. Figure 12 only shows 2 points closer to the turbine x = 0.5D and not x = 2D.**

Thank you for this comment. In the final version, we opted to only show the points at x=0.5D. These two points give insight in the signal content in the frequency domain, which is further described in the text. We decided not to show the farther downstream points to reduce the length of the paper. However, the other two points are covered in the dissertation of one of our authors (Florian Heckmeier, mediaTUM - Medien- und Publikationsserver). We changed the text accordingly.

As a next step, in order to discover the governing mechanisms in the actuated cases, kinetic energy spectra are calculated for various locations in the wake of the turbine. Figure 11 shows the spectra at  two locations $P1 = (x/D, y/D) = (0.5, -0.55)$, and $P2 = (0.5, -0.15)$.
450 As expected, the baseline spectra solely experience peaks at the rotational frequency $f_r = 14\,Hz$ and its higher harmonics.  In the more inward locations inside the wake at

**10) Shifting the spectra in figure 12 vertically could help to better show the peaks for the different conditions.**

Thank you for pointing this out. We adapted the y-axis range for both plots to better show the peaks:

[Figure]

(a) $x/D = 0.5$ and $y/D = -0.55$          (b) $x/D = 0.5$ and $y/D = -0.15$

**Figure 11.** Kinetic energy spectra $E_{kin}$ at various measurement locations in the turbine wake at $x/D = 0.5$ and $y/D = \{-0.15, -0.55\}$.

**11) Line 465 second sentence: Why does the measured data solely consist of the azimuthal and the blade 1 pitch position? Why blade 1? Even though the data is phase-locked analysed, the resulting wake is a results of the complete rotor. Am I wrong?**

Thank you for this question. Unfortunately, we think this is a misunderstanding. The acquired data has two entries for blade 1, one for the azimuthal position and one for the blade pitch. The data for the other blades can be deduced by a) the geometric properties of the 3 bladed turbine and b) by the actuation law as introduced in the theoretical part. We hope that by this answer, we can clarify your concerns.

**12) In section 4.2.3 the authors explain how they got the data for the phase-locked data with the additional frequency. I think it could help to show graphically how they get the beat frequency and how they apply this to the five hole data.**

Thank you for this remark. We added some explanation and a figure in section 4.2.3 showing how the envelope/beat frequency signal is extracted from the blade azimuthal and pitch position.

a sinusoidal signal with $f_e = 2.5\ Hz$ that is used in the following to phase-lock the measured FRAP data. The process of extracting the envelope/beat frequency from the blade azimuth and pitch position is visualized in Figure 17. After extracting the time series for the beat frequency for each individual measurement point, the measured FRAP data are phase-averaged with the beat frequency. Readings of all measurement points can thereby be correlated.

[Figure]

**Figure 17.** Schematic visualization of the process of extracting the envelope/beat frequency from the blade azimuthal and pitch position for phase-locking with the additional frequency.

13) **For this analysis, how did you make sure that the pitching starts at the same position of the blades?**

Thank you for this question. We added a paragraph, in section 3.3, that discusses about this aspect.

230 During each experiment, the Helix was activated a few seconds before the start of the acquisition, to allow for the downstream propagation of the new wake. The activation of the Helix, however, was not synchronized to a specific azimuth position of the rotor. For $f_\beta \neq f_r$ this aspect is not relevant and does not affect the results. In this regard, each plot in Fig. 3 shows, for 5 of the conducted experiments, the distribution of the first blade azimuth ($\theta_1$) detected when the required pitch is within the range $0.98\beta_{max} < \beta_1 < 1.02\beta_{max}$, with $\beta_{max} = \hat{\beta} + \beta_0 = 4° + 0.4°$ the maximum requested pitch

235 angle. The dashed black line, instead, marks the azimuth position ($\theta_{1,ini}$) at the very first time the required pitch is within the range $0.98\beta_{max} < \beta_1 < 1.02\beta_{max}$. For experiments conducted with $f_\beta \neq f_r$, the pitch is detected around $\beta_{max}$ for the whole range of azimuth positions (the distribution is not completely homogeneous solely due to the discrete sampling), regardless of the value of $\theta_{1,ini}$. This implies that the fixed-frame moments produced by the Helix rotate over the entire range of azimuth positions, which leads to the expected effects on the wake.

240 Different is the case with $f_\beta = f_r$; the maximum pitch is indeed always detected at the same azimuth position, i.e. around $\theta_{ini}$. Previous experimental (Campagnolo et al., 2016) and numerical (Fleming et al., 2014) investigations found that the azimuth position of maximum pitch affects the achieved amount of wake deflection. However, the study of Wang et al. (2016) revealed that the power gains observed on the downstream machine may only be a little higher than the power losses experienced by the upstream machine, at the price of a significantly increased loading of the upstream machine. In this article, it was therefore

245 preferred to exclude the case $f_\beta = f_r$ from the following discussions. A complete analysis would, indeed, require investigating

[Figure]

**Figure 3.** Distributions of the first blade azimuth ($\theta_1$) detected when the required pitch is close to $\beta_{max}$, and with $f_\beta/f_r = [0.72, 0.86, 1.12, 1.28, 1]$. The dashed black lines mark $\theta_{1,ini}$, i.e. the azimuth position recorded at the very first time the required pitch is within the range $0.98\beta_{max} < \beta_1 < 1.02\beta_{max}$.

**14) Is there any reason, why the authors do not perform the same analysis with the fluctuations for the phase-locked with additional frequency?**

Thank you for this question. We also analysed the streamwise fluctuations $u'_{rms}$. However, the results of this analysis did not reveal additional information to the ones of the vorticity. As you can see in Figure 1, where the streamwise fluctuations for the Helix case 1.18 are presented, the pronounced area is slightly wider. Still, they show the same meandering as the ones of the vorticity, presented in the paper. As we did not want to further extend the size of the article, we did not consider this analysis adding new information. Consequently, we decided not to show these plots and present an analysis. We added a sentence in section 4.2.3.

[Figure]

*Figure 1. Streamwise fluctuations $u'_{rms}/U_\infty$ for the Helix 1.18 case.*

which are representative  of the phase-locking analysis with the  azimuth position. An analysis of the streamwise fluctuations $u'_{rms}/U_\infty$ phase-locked with the additional frequency $f_e$ resulted in similar plots, showing the same
560   meandering motion and are for this reason not presented here. The figures show snapshots of multiple phase-locked positions

**15) The authors state that the meandering of the tip vortices in the blade tip region is the main driver for the increase wake mixing. Following their analysis, I can only conclude that there is a meandering of the blade tip vortices. By performing the same analysis for other St_add, for wich the total power is not or just slightly increased, should provide clear evidence that this meandering is either not there or decreased. The authors should add such an analysis.**

Thank you for pointing out this. We agree that such an analysis would be very interesting and help clarify if the meandering is the main driver. When we conducted the experimental campaign, we decided to perform detailed wake measurements for the conditions where the highest effects on the wake were expected. By doing this, we hoped we could explain the mechanisms best. After such a study and getting more insight into the topic, having data for other frequencies would also be interesting. Unfortunately, it is impossible to perform another experimental analysis on this. Nevertheless, we are working on an article investigating the Helix technique for a 2-bladed rotor numerically. Figure 2 shows results of the wake, for different frequencies for CW and CCW rotation. For $f_\theta/f_r$ = 0.82 and $f_\theta/f_r$ = 1.18, which are similar to the optimum cases of this article, we see significantly stronger wake meandering than $f_\theta/f_r$ = 0.9 and $f_\theta/f_r$ = 1.10. We hope this helps to confirm our assumption and clarify your concerns. Nevertheless, as this analysis is not part of our article, we weakened the statement slightly. We wrote "…the meandering of the vortex in the blade tip region is expected to be the main driver for wake mixing…" instead of "…the meandering of the vortex in the blade tip region is the main driver for wake mixing…". Furthermore, the sentence was added to the conclusions and outlook.

[Figure]

*Figure 2. Results from simulations for a 2-bladed rotor operated with the Helix technique for different St_add*

> Nevertheless, the Helix technique and its real-world realization  are just at the beginning of its development. Consequently, this study can only provide a first insight  into the potential and wake mixing mechanisms.  Additional wake analysis with different Stroual numbers is required to prove that the wake meandering is the main driver for increased mixing. Moreover, an
> 665 investigation of the influence of inflow turbulence on  wake mixing is needed to understand its potential in more realistic inflow conditions characterized by higher levels of ambient turbulence. Further, testing the Helix technique in wind farm control studies could be promising.

**Technical comments**

**The authors mix the value for the Strouhal number St_add — that value is sometimes 0.45 and sometimes 0.47. According to the definition it should be 0.47 all the time unless I missed something.**

Thank you for this hint. We changed this throughout the full text to 0.47, which is, as you mentioned, the correct value following the definition of St_add.

**Line 308: as seen in figure 8**

Thank you for this comment. We corrected this typo.

**Line 368: what does the index „1" stand for in the definition of P1 = (x/D, y/D)_1 ??**

Thank you for this comment. We intended to add this index to show that this is the x and y coordinates of point 1. However, since this could lead to misunderstandings, we removed it.

**Line 411, last sentence should be x=5D for the complete merge of the tip vortices.**

Thank you for this comment. We totally agree on that and changed it accordingly.

**Line 439: the youtube link is not finalised.**

Thank you for point this out, we added the final link.

**Line 443: Isn't the „"additional rotational frequency f_(r,a) identical to the additional excitation frequency f_e ?**

Thank you for hinting at this mistake. Throughout our studies, we used several nomenclatures and this is a**n** erroneous remnant. We changed it to f_e.

**Line 474: Youtube link is not finalised.**

Thank you for point this out, we added the final link.

**Line 493: in which an x-y plane**

Thank you for this comment. We added the missing "n".

---

## Author Comment (AC3)

**Authors' response to Referee 3**

**General**

**The authors present an experimental study in a wind tunnel for a control strategy for wind turbines. The control system, named Helix, increases the rotational component of the wake by pitch control for sinusoidally varying yaw and tilt moments. Experiments are performed under low turbulence conditions using a single or two scaled turbines, studying in different steps, the wake averaged statistics and phase-locking techniques. Also, several sensors provide turbine level observations. The authors then discuss and quantify wake recovery and vortex meandering. It is found that operating the scaled turbine with the Helix control results in faster wake recovery when compared with the baseline cases.**
**The manuscript is well written and the experiments and results of interest for the wind energy community. Nevertheless, before recommending publications I ask the authors to address the following comments and remarks:**

We thank the referee for reviewing this manuscript, the valuable feedback, and the constructive comments. At this stage of the review process, we respond to referee #3's comments and propose improvements for the journal manuscript. The referee's original comments are printed in bold followed by the corresponding answers. A screenshot of the different versions of the updated passages from the manuscript is provided below the answer.

**Specific comments**

1) **The study concerns low turbulence conditions only. Nevertheless, background turbulence significantly affects the development of the wake and the structures within it. This therefore raises the question, discussed by the authors in the introduction, about the relevance of present results in realistic conditions. While the present study presents a fundamental interest, I consider that the authors should discuss in better detail, using the several works available in the literature, how their results will be modified when background turbulence is present.**

Thank you for raising this point. The same topic was also brought up by referee #2. We agree that it is important to address this point better. We added a small discussion about this in the literature review in the introduction chapter. We added a source of a study where the authors investigate the effect of inflow turbulence on the efficiency of dynamic wake mixing and show that inflow turbulence has a significant inflow on the effectiveness of wake mixing for power optimization. Furthermore, we updated the future works slightly to say that further investigations on inflow turbulence are needed.

> approach experimentally in a wind tunnel (W/T) and should give a detailed insight in the wake aerodynamics. To provide
> 60 such a detailed insight, the flow in the wind tunnel has to be a clean lab flow, which is uniform and is characterized by a very small turbulence intensity. Such clean inflow will not only highlight the effects of the control technique in the wake but also influence their effectiveness. Wake mixing techniques like Helix add turbulence to the wake. Consequently, if the turbine inflow is already characterized by higher ambient turbulence the effect of wake mixing will be mitigated. In a recent study Mühle et al. (2024) compare power gains for wake mixing by dynamic yaw for different inflow turbulence. They found
> 65 a strong reduction of the effect on the power of a two-turbine setup in case of high inflow turbulence and thus confirm the findings of Munters and Meyers (2018a). Nevertheless, they suggest that wake mixing has the potential to improve the power output of a wind farm in case the wakes are strong and persistent.

> Nevertheless, the Helix technique and its real-world realization  are just at the beginning of its development. Consequently, this study can only provide a first insight  into the potential and wake mixing mechanisms.  Additional wake analysis with different Stroual numbers is required to prove that the wake meandering is the main driver for increased mixing. Moreover, an
> 665 investigation of the influence of inflow turbulence on  wake mixing is needed to understand its potential in more realistic inflow conditions characterized by higher levels of ambient turbulence. Further, testing the Helix technique in wind farm control studies could be promising.

**2) Also, the setup presents a large blockage. This is also briefly discussed by the authors, and they use a very simple model to cater for this issue. Nevertheless, blockage not only affects the hub velocity but it also severely modifies the wake development, the air it entrains and the evolution of structures. Several works discuss the relevance of blockage and propose different corrections (see for instance Saghi et al 2016, Steiros et al 2022, among several others). Blockage is one of the main limitations of the experimental setup and should be addressed carefully.**

Thank you for this important comment. The same concern was also raised by Reviewer #1 and Reviewer #2. We agree that the blockage is very high and was not discussed appropriately. We added information about the blockage effect in section 3.2. We included a paragraph in which we use several studies investigating the blockage effect to discuss the effect that blockage is expected to have on wake development.

> however accelerated, resulting in a higher velocity experienced by the turbines.  To account for such an
> 180 effect on the turbine performance, it can be corrected by applying analytical models, a recent review is presented in the study by Steiros et al. (2022). More information is presented by Ross and Polagye (2020) who conducted an experimental assessment of such models and their application to different wind turbine concepts. In the present study, the performance was corrected by a calculation of the Rotor Effective Wind Speed (REWS) for the upstream G1, done as described in Campagnolo et al. (2022). This revealed a blockage-corrected free-stream velocity $U_{\infty,corr} \approx 5.9\ m/s$, which correlates to the rated wind speed
> 185 of the model wind turbine. Consequently, the turbine is operated at a tip-speed ratio of approx. $\lambda = 8.2$.  Based on the authors' knowledge, there are no analytical blockage correction models that would allow us to quickly assess the influence of the wind tunnel walls on the wake. One possible way to investigate this is to use computational fluid dynamics (CFD). Zaghi et al. (2016) studied the effect of blockage on a model wind turbine with a Reynolds Averaged Navier-Stokes (RANS) simulation. They analyzed the streamwise wake velocity and found increased velocities in the area behind the rotor but also
> 190 in the outer region of the wake in case a wind tunnel wall was present. In a more detailed analysis Sarlak et al. (2016) used Large Eddy Simulation (LES) combined with the actuator line technique to investigate wake velocity and Reynolds stresses for different blockage ratios up to $\alpha = 0.2$. They found a significant impact on the mean wake velocity in the case of the highest blockage ratio. Especially in the region outside of the rotor, the velocity is found to increase; this augmentation is mitigated in the rotor area but is still present. Furthermore, they concluded that blockage has no considerable effect on the
> 195 wake mixing rate as maximum and minimum velocities do not differ significantly. In a combined experimental and numerical study McTavish et al. (2014) investigated the influence of wind tunnel blockage on the wake width and found that the wake compresses when blockage increases. Consequently, the wake results of the presented study are expected to be characterized by slightly higher streamwise velocities and a narrower wake compared to a full-scale test. As a result, in the analysis of the

**3) The time-resolved five-hole probe has a large head surface (around 8 squared millimeters) and therefore, for a turbulent wake in a wind tunnel, lies within the inertial range of turbulence. It is then important to check the effective temporal resolution, taking into account both background noise in the wind tunnel and spatial filtering effects. I therefore suggest that the authors show some typical spectra obtained with the probe. Also, the description of the calibration process is quite long, has it been performed by the authors or by the manufacturer? If it is the latter case, I suggest that the discussion is taken out of the manuscript.**

Thank you for your comment on the probe. Since Reviewer 1 also asked for changes related to the FRAP, we changed the text. We removed the detailed section on the FRAP, despite the probe being developed and calibrated by the author in collaboration with the manufacturer (see Dissertation of F.M. Heckmeier). To answer your question on the spatial and temporal resolution of the probe, we would like to refer to a study we performed targeting this question. In this study, we addressed this topic and compared the probe to hot-wire probes using grid-generated turbulence ((2) (PDF) Spatial and temporal resolution of a fast-response aerodynamic pressure probe in grid-generated turbulence (researchgate.net)).

We ensured the appropriate FRAP usage and showed the probe's spatial and temporal limitations. We added this reference to the text (see also response to Reviewer 1, Question 5).

> The spatial and temporal characteristics and the underlying calibration process of the FRAP can be found in the literature
> 265   (see Heckmeier et al. (2019); Heckmeier and Breitsamter (2020); Heckmeier et al. (2021); Heckmeier (2022)): A high spatial
> accuracy below $0.2°$ in both flow angles and $0.1 \, m/s$ in the reconstructed velocity can be achieved. The spatial and temporal

[Figure]

**Figure 4.** Schematic fast-response five-hole probe and G1 wind turbine model measurement setup for TUM-AER W/T-C

> resolution of the applied FRAP has been investigated and hence, shows the suitability of the usage of the FRAP for this
> experiment.

**4) Figure 10 suggests, despite the presence of an adjustable ceiling, a significant pressure gradient in the tunnel. Is that the case or an effect of the y-axis limit?**

Thank you for this question. I think this is a misunderstanding and due to the figure limits of the horizontal/y-axis. The wind tunnel has a width of 2.7m. The y-axis is limited by 0.85D=0.935m. Hence, there is an additional ca. 40cm distance from the measurement location to the wind tunnel wall (see the red line in the figure below). We hope this clarifies your question.

[Figure]

5) **In its current form, the manuscript is very long and, given the large number of results presented, some sections are hard to follow. The authors should consider putting some results and discussions in an appendix.**

Thank you for the hint. The manuscript is indeed quite long; unfortunately, the effects of the Helix are complex and, so far, not deeply investigated. We think all the results, figures, and discussions reported in the paper are needed, to provide a complete picture of the method and its impacts.

**Technical comments**

**The manuscript is overall very well written, but it still presents several typos.**

Thank you for pointing this out. Yes, also reviewers #1 and #2 have pointed out several typos. We covered all of these, so the manuscript should be in a good state now.